# SIMPLE EMERGENT ACTION REPRESENTATIONS FROM MULTI-TASK POLICY TRAINING

**Pu Hua[1,4], Yubei Chen[*2], Huazhe Xu[*1,3,4]**
[1]Tsinghua University, [2]Center for Data Science, New York University, [3]Shanghai AI Lab,
[4]Shanghai Qi Zhi Institute

## ABSTRACT

The low-level sensory and motor signals in deep reinforcement learning, which exist in high-dimensional spaces such as image observations or motor torques, are inherently challenging to understand or utilize directly for downstream tasks. While sensory representations have been extensively studied, the representations of motor actions are still an area of active exploration. Our work reveals that a space containing meaningful action representations emerges when a multi-task policy network takes as inputs both states and task embeddings. Moderate constraints are added to improve its representation ability. Therefore, interpolated or composed embeddings can function as a high-level interface within this space, providing instructions to the agent for executing meaningful action sequences. Empirical results demonstrate that the proposed action representations are effective for intra-action interpolation and inter-action composition with limited or no additional learning. Furthermore, our approach exhibits superior task adaptation ability compared to strong baselines in Mujoco locomotion tasks. Our work sheds light on the promising direction of learning action representations for efficient, adaptable, and composable RL, forming the basis of abstract action planning and the understanding of motor signal space. Project page: `https://sites.google.com/view/emergent-action-representation/`

## 1 INTRODUCTION

Deep reinforcement learning (RL) has shown great success in learning near-optimal policies for performing low-level actions with pre-defined reward functions. However, reusing this learned knowledge to efficiently accomplish new tasks remains challenging. In contrast, humans naturally summarize low-level muscle movements into high-level action representations, such as "pick up" or "turn left", which can be reused in novel tasks with slight modifications. As a result, we carry out the most complicated movements without thinking about the detailed joint motions or muscle contractions, relying instead on high-level action representations (Kandel et al., 2021). By analogy with such abilities of humans, we ask the question: can RL agents have action representations of low-level motor controls, which can be reused, modified, or composed to perform new tasks?

As pointed out in Kandel et al. (2021), "the task of the motor systems is the reverse of the task of the sensory systems. Sensory processing generates an internal representation in the brain of the outside world or of the state of the body. Motor processing begins with an internal representation: the desired purpose of movement." In the past decade, representation learning has made significant progress in representing high-dimensional sensory signals, such as images and audio, to reveal the geometric and semantic structures hidden in raw signals (Bengio et al., 2013; Chen et al., 2018; Kornblith et al., 2019; Chen et al., 2020; Baevski et al., 2020; Radford et al., 2021; Bardes et al., 2021; Bommasani et al., 2021; He et al., 2022; Chen et al., 2022). With the generalization ability of sensory representation learning, downstream control tasks can be accomplished efficiently, as shown by recent studies Nair et al. (2022); Xiao et al. (2022); Yuan et al. (2022). While there have been significant advances in sensory representation learning, action representation learning remains largely unexplored. To address this gap, we aim to investigate the topic and discover generalizable action representations that can be reused or efficiently adapted to perform new tasks. An important concept

---

[*]Denotes equal contributions.

in sensory representation learning is pretraining with a comprehensive task or set of tasks, followed by reusing the resulting latent representation. We plan to extend this approach to action representation learning and explore its potential for enhancing the efficiency and adaptability of reinforcement learning agents. We propose a multi-task policy network that enables a set of tasks to share the same latent action representation space. Further, the time-variant sensory representations and time-invariant action representations are decoupled and then concatenated as the sensory-action representations, which is finally transformed by a policy network to form the low-level action control. Surprisingly, when trained on a comprehensive set of tasks, this simple structure learns an emergent self-organized action representation that can be reused for various downstream tasks. In particular, we demonstrate the efficacy of this representation in Mujoco locomotion environments, showing zero-shot interpolation/composition and few-shot task adaptation in the representation space, outperforming strong meta RL baselines. Additionally, we find that the decoupled time-variant sensory representation exhibits equivariant properties. The evidence elucidates that reusable and generalizable action representations may lead to efficient, adaptable, and composable RL, thus forming the basis of abstract action planning and understanding motor signal space. The primary contributions in this work are listed as follows:

1. We put forward the idea of leveraging emergent action representations from multi-task learners to better understand motor action space and accomplish task generalization.

2. We decouple the state-related and task-related information of the sensory-action representations and reuse them to conduct action planning more efficiently.

3. Our approach is a strong adapter, which achieves higher rewards with fewer steps than strong meta RL baselines when adapting to new tasks.

4. Our approach supports intra-action interpolation as well as inter-action composition by modifying and composing the learned action representations.

Next, we begin our technical discussion right below and leave the discussion of many valuable and related literature to the end.

## 2  PRELIMINARIES

**Soft Actor-Critic.**   In this paper, our approach is built on Soft Actor-Critic (SAC) (Haarnoja et al., 2018). SAC is a stable off-policy actor-critic algorithm based on the maximum entropy reinforcement learning framework, in which the actor maximizes both the returns and the entropy. We leave more details of SAC in Appendix A.

**Task Distribution.**   We assume the tasks that the agent may meet are drawn from a pre-defined task distribution $p(\mathcal{T})$. Each task in $p(\mathcal{T})$ corresponds to a Markov Decision Process (MDP). Therefore, a task $\mathcal{T}$ can be defined by a tuple $(\mathcal{S}, \mathcal{A}, P, p_0, R)$, in which $\mathcal{S}$ and $\mathcal{A}$ are respectively the state and action space, $P$ the transition probability, $p_0$ the initial state distribution and $R$ the reward function.

The concept of task distribution is frequently employed in meta RL problems, but we have made some extensions on it to better match with the setting in this work. We divide all the task distributions into two main categories, the "uni-modal" task distributions and the "multi-modal" task distributions. Concretely, the two scenarios are defined as follows:

- *Definition 1 (Uni-modal task distribution):* In a uni-modal task distribution, there is only one modality among all the tasks in the task distribution. For example, in HalfCheetah-Vel, a Mujoco locomotion environment, we train the agent to run at different target velocities. Therefore, running is the only modality in this task distribution.

- *Definition 2 (Multi-modal task distribution):* In contrast to uni-modal task distribution, there are multiple modalities among the tasks in this task distribution. A multi-modal task distribution includes tasks of several different uni-modal task distributions. For instance, we design a multi-modal task distribution called HalfCheetah-Run-Jump, which contains two modalities including HalfCheetah-BackVel and HalfCheetah-BackJump. The former has been defined in the previous section, and the latter contains tasks that train the agent to jump with different reward weight. In our implementation, we actually train four motions in this environment, running, walking, jumping ans standing. We will leave more details in Section 4 and Appendix B.1.

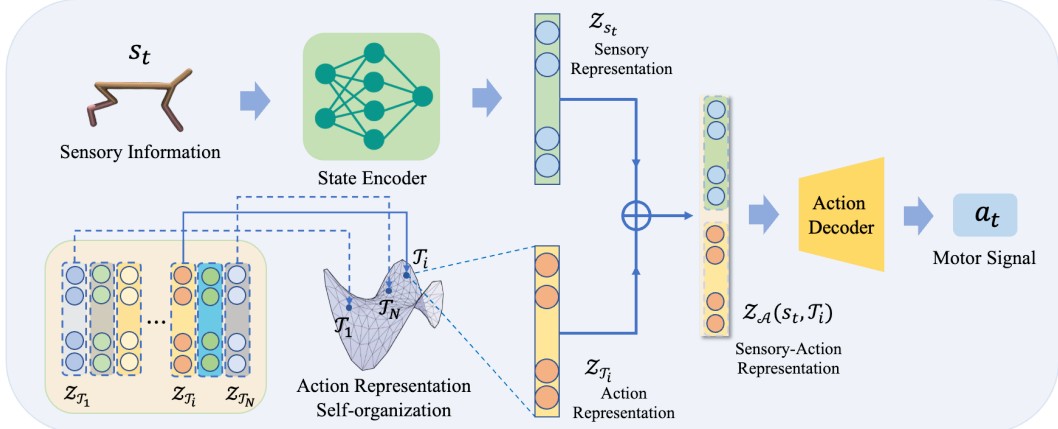

Figure 1: **Emergent action representations from multi-task training.** The sensory information and task information are encoded separately. When both are concatenated, an action decoder decodes them into a low level action.

# 3 EMERGENT ACTION REPRESENTATIONS FROM MULTI-TASK TRAINING

In this section, we first introduce the sensory-action decoupled policy network architecture. Next, we discuss the multitask policy training details, along with the additional constraints to the task embedding for the emergence of action representations. Lastly, we demonstrate the emergence of action representations through various phenomena and applications.

## 3.1 MULTITASK POLICY NETWORK AND TRAINING

**Decoupled embedding and concatenated decoding.** An abstract high-level task, e.g., "move forward", typically changes relatively slower than the transient sensory states. As a simplification, we decouple the latent representation into a time-variant sensory embedding $\mathcal{Z}_{\mathbf{s}_t}$ and a time-invariant task embedding $\mathcal{Z}_{\mathcal{T}}$, which is shown in Figure 1. These embeddings concatenate to form a sensory-action embedding $\mathcal{Z}_{\mathcal{A}}(\mathbf{s}_t, \mathcal{T}) = [\mathcal{Z}_{\mathbf{s}_t}, \mathcal{Z}_{\mathcal{T}}]$, which is transformed by the policy network (action decoder) $\psi$ to output a low-level action distribution $p(\mathbf{a}_t) = \psi(\mathbf{a}_t | \mathcal{Z}_{\mathbf{s}_t}, \mathcal{Z}_{\mathcal{T}})$, e.g., motor torques. The action decoder $\psi$ is a multi-layer perceptron (MLP) that outputs a Gaussian distribution in the low-level action space $\mathcal{A}$.

**Latent sensory embedding (LSE).** The low-level sensory state information is encoded by an MLP state encoder $\phi$ into a latent sensory embedding $\mathcal{Z}_{\mathbf{s}_t} = \phi(\mathbf{s}_t) \in \mathbb{R}^m$. It includes the proprioceptive information of each time step. LSE is time-variant in an RL trajectory, and the state encoder is shared among different tasks. We use LSE and sensory representation interchangeably in this paper.

**Latent task embedding (LTE).** A latent task embedding $\mathcal{Z}_{\mathcal{T}} \in \mathbb{R}^d$ encodes the time-invariant knowledge of a specific task. Let's assume we are going to train $N$ different tasks, and their embeddings form an LTE set $\{\mathcal{Z}_{\mathcal{T}_N}\}$. These $N$ different tasks share the same state encoder $\phi$ and action decoder $\psi$; in other words, these $N$ tasks share the same policy network interface, except for their task embeddings being different. For implementation, we adopt a fully-connected encoder, which takes as input the one-hot encodings of different training tasks, to initialize the set $\{\mathcal{Z}_{\mathcal{T}_N}\}$. This task encoder is learnable during training.

After training, the LTE interface can be reused as a high-level action interface. Hence, we use LTE and action representation interchangeably in this paper.

**Training of the multi-task policy networks.** A detailed description of the multi-task training is demonstrated in Algorithm 1. When computing objectives and their gradients, we use policy $\pi$ parameterized by $\omega$ to indicate all the parameters in the state encoder, action decoder, and $\{\mathcal{Z}_{\mathcal{T}_N}\}$. The overall training procedure is based on SAC. The only difference is that the policy network and Q networks additionally take as input the LTE $\mathcal{Z}_{\mathcal{T}}$ and a task label, respectively. During training,

we also apply two techniques to constrain this space: 1) we normalize the LTEs so that they lie on a hypersphere; 2) we inject a random noise to the LTEs to enhance the smoothness of the space. An ablation study on these two constraints is included in Appendix B.7.

---

**Algorithm 1** Multi-task Training

---

**Input:** Training task set $\{\mathcal{T}_N\} \sim p(\mathcal{T}), \theta_1, \theta_2, \omega$
    $\bar{\theta}_1 \leftarrow \theta_1, \bar{\theta}_2 \leftarrow \theta_2, \mathcal{B} \leftarrow \emptyset$
    Initialize LTE set $\{\mathcal{Z}_{\mathcal{T}_N}\}$ for $\{\mathcal{T}_N\}$
    **for** each pre-train epoch **do**
        **for** $\mathcal{T}_i$ in $\{\mathcal{T}_n\}$ **do**
            Sample a batch $\mathcal{B}_i$ of multi-task RL transitions with $\pi_\omega$
            $\mathcal{B} \leftarrow \mathcal{B} \cup \mathcal{B}_i$
        **end for**
    **end for**
    **for** each train epoch **do**
        Sample RL batch $b \sim \mathcal{B}$
        **for all** transition data in $b$ **do**
            $\mathcal{Z}_{\mathbf{s}_t} = \phi(\mathbf{s}_t)$
            $\widetilde{\mathcal{Z}}_{\mathcal{T}_i} = \text{normalize}(\mathcal{Z}_{\mathcal{T}_i} + n)$ and $n \sim \mathcal{N}(0, \sigma^2)$
            Sample action $\mathbf{a}_t \sim \psi(\cdot | \mathcal{Z}_{\mathbf{s}_t}, \widetilde{\mathcal{Z}}_{\mathcal{T}_i})$ for computing SAC objectives
        **end for**
        **for** each optimization step **do**
            Compute SAC objectives $J(\alpha), J_\pi(\omega), J_Q(\theta)$ with $b$ based on Equation 234
            Update SAC parameters
        **end for**
    **end for**
**Output:** The optimal model of state encoder $\phi^*$ and action decoder $\psi^*$ and a set of LTEs $\{\mathcal{Z}_{\mathcal{T}_N}\}$

---

## 3.2 THE EMERGENCE OF ACTION REPRESENTATION

After we train the multi-task policy network with a comprehensive set of tasks, where the LTE vectors in $\{\mathcal{Z}_{\mathcal{T}_N}\}$ share the same embedding space, we find that $\{\mathcal{Z}_{\mathcal{T}_N}\}$ self-organizes into a geometrically and semantically meaningful structure. Tasks with the same modality are embedded in a continuous fashion, which facilitates intra-task interpolation. Surprisingly, the composition of task embeddings from different modalities leads to novel tasks, e.g., "run" + "jump" = "jump run". Further, the action representation can be used for efficient task adaptation. Visualization also reveals interesting geometric structures in task embedding and sensory representation spaces. In this subsection, we dive into these intriguing phenomena, demonstrating the emergence of action representation and showing the generalization of the emergent action representation.

**Task interpolation & composition.** After training the RL agent to accomplish multiple tasks, we select two pre-trained tasks and generate a new LTE through linear integration between the LTEs of the two chosen tasks. The newly-generated task embedding is expected to conduct the agent to perform another different task. The generated LTE is defined by:

$$\mathcal{Z}' = f(\beta \mathcal{Z}_{\mathcal{T}_i} + (1 - \beta)\mathcal{Z}_{\mathcal{T}_j}) \tag{1}$$

where $i, j$ are the indices of the selected tasks and $\mathcal{Z}_{\mathcal{T}_i}, \mathcal{Z}_{\mathcal{T}_j}$ are their corresponding LTEs. $\beta$ is a hyperparameter ranging in (0,1). The function $f(\cdot)$ is a regularization function related to the pre-defined quality of the LTE Space. For instance, in this paper, $f(\cdot)$ is a normalization function to extend or shorten the result of interpolation to a unit sphere.

A new task is interpolated by applying the aforementioned operation on the LTEs of tasks sampled from a uni-modal distribution. The interpolated task usually has the same semantic meaning as the source tasks while having different quantity in specific parameters, e.g., running with different speeds. A new task is composed by applying the same operation on tasks sampled from a multi-modal distribution. The newly composed task usually lies in a new modality between the source

tasks. For example, when we compose "run" and "jump" together, we will have a combination of an agent running while trying to jump.

**Efficient adaptation.** We find that an agent trained with the multi-task policy network can adapt to unseen tasks quickly by only optimizing the LTEs. This shows that the LTEs learn a general pattern of the overall task distribution. When given a new task after pre-training, the agent explores in the LTE Space to find a suitable LTE for the task. Specifically, we perform a gradient-free cross-entropy method (CEM) (De Boer et al., 2005) in the LTE space for accomplishing the desired task. Detailed description can be found in Algorithm 2.

**Geometric Structures of LTEs and the LSEs.** We then explore what the sensory representations and the action representation space look like after multi-task training. In order to understand their geometric structures, we visualize the LSEs and LTEs. The detailed results of our analysis are presented in Section 4.5.

---

**Algorithm 2** Adaptation via LTE Optimization

---

**Input:** Adaptation task $\mathcal{T} \sim p(\mathcal{T})$, $\phi^*, \psi^*$, capacity of elite set $m$, number of sampling $n$
  Initialize the elite set $\mathbb{Z}_e$ with $m$ randomly sampled LTEs from the LTE Space
  **for** each adapt epoch **do**
    Initialize the overall test set by $\mathbb{Z} \leftarrow \emptyset$
    **for** $\mathcal{Z}_i$ in $\mathbb{Z}_e$ **do**
      Sample $n$ LTEs $\mathcal{Z}_{i1}, \ldots, \mathcal{Z}_{in}$ near $\mathcal{Z}_i$
      $\mathbb{Z} \leftarrow \mathbb{Z} \cup \{\mathcal{Z}_i, \mathcal{Z}_{i1}, \ldots, \mathcal{Z}_{in}\}$
    **end for**
    **for** $\mathcal{Z}_j$ in $\mathbb{Z}$ **do**
      **while** not done **do**
        $\mathcal{Z}_{\mathbf{s}_t} = \phi^*(\mathbf{s}_t),$
        $\mathbf{a}_t \sim \psi^*(\cdot|\mathcal{Z}_{\mathbf{s}_t}, \mathcal{Z}_j)$
        $r_t = R(\mathbf{s}_t, \mathbf{a}_t|\mathcal{T})$
        $\mathbf{s}_{t+1} \sim p(\mathbf{s}_{t+1}|\mathbf{s}_t, \mathbf{a}_t)$
      **end while**
    **end for**
    Sort the task embeddings in $\mathcal{Z}$ by high cumulative reward in the trajectory
    Select the top $m$ LTEs in $\mathcal{Z}$ to update $\mathbb{Z}_e$
  **end for**

---

## 4 EXPERIMENTS

In this section, we first demonstrate the training process and performance of the multi-task policy network. Then, we use the LTEs as a high-level action interface to instruct the agents to perform unseen skills through interpolation without any training. After that, we conduct experiments to evaluate the effectiveness of the LTEs in task adaptation. Lastly, we visualize the LSEs and LTEs to further understand the structure of the state and action representation. We use **e**mergent **a**ction **r**epresentation (EAR) to refer to the policy using the LTEs.

### 4.1 EXPERIMENTAL SETUPS

**Environments.** We evaluate our method on five locomotion control environments (HalfCheetah-Vel, Ant-Dir, Hopper-Vel, Walker-Vel, HalfCheetah-Run-Jump) based on OpenAI Gym and the Mujoco simulator. Detailed descriptions of these RL benchmarks are listed in Appendix B.1. Beyond the locomotion domain, we also conduct a simple test on our method in the domain of robot manipulation, please refer to Appendix B.6 for detailed results.

**Baselines.** We compare EAR-SAC, the emergent action representation based SAC with several multi-task RL and meta RL baselines. For multi-task RL baselines, we use multi-head multi-task SAC (MHMT-SAC) and one-hot embedding SAC (OHE-SAC; for ablation). For meta RL baselines, we use MAML (Finn et al., 2017) and PEARL (Rakelly et al., 2019). Detailed descriptions of these baselines are listed in Appendix B.2.

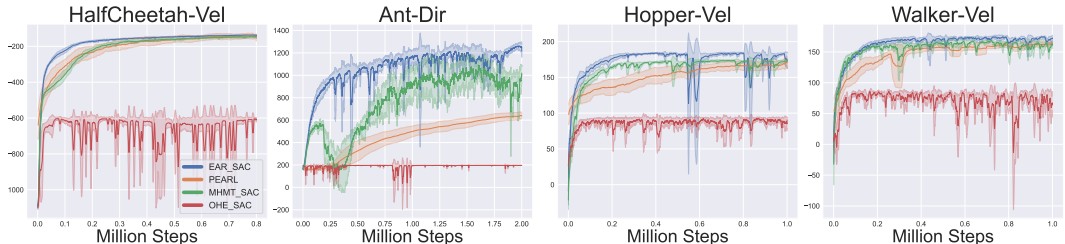

Figure 2: **Training performance of our method and baselines.** The x-axis represents the total training steps (in million steps) and the y-axis represents the average reward of all the training tasks. All the training are based on 3 seeds. The shaded area is one standard deviation.

| Method | HalfCheetah-Vel | Ant-Dir | Hopper-Vel | Walker-Vel |
|---|---|---|---|---|
| **EAR-SAC (Ours)** | **-136.9±1.29** | **1216.7±54.97** | **173.5±11.09** | **172.7±4.05** |
| MAML | -500.9±47.33 | 422.8±29.40 | 34.0±41.92 | -4.9±0.58 |
| PEARL | -155.0±15.50 | 635.9±19.21 | 170.0±7.80 | 163.9±1.07 |
| MHMT-SAC | -145.2±2.60 | 1020.0±54.90 | 172.9±1.17 | 160.6±5.21 |
| OHE-SAC | -609.0±8.10 | 195.7±0.29 | 90.7±2.39 | 65.0±20.83 |

Table 1: **Comparison with baselines on the final performance.** The metric is the return of the last epoch, and the mean and standard deviation is calculated among 3 seeds.

## 4.2 MULTI-TASK TRAINING FOR ACTION REPRESENTATIONS

In this section, we train the multi-task network and evaluate whether sensory-action representations can boost training efficiency. EAR-SAC is compared with all the multi-task RL and meta RL baselines on final rewards in Table 1 and additionally compared with off-policy baselines on the training efficiency in Figure 2. We find that EAR-SAC outperforms all the baselines in terms of training efficiency. In environments with high-dimensional observations (e.g., Ant-Dir), EAR-SAC achieves large performance advantage against the baselines. We attribute this to that the learned action representation space may provide meaningful priors when the policy is trained with multiple tasks.

## 4.3 ACTION REPRESENTATION AS A HIGH-LEVEL CONTROL INTERFACE

In this section, we control the agent by recomposing the LTEs. We perform intra-action interpolation and inter-action composition in uni-modal task distributions and multi-modal task distributions respectively.

**Intra-action interpolation.** Intra-action interpolation is conducted in HalfCheetah-Vel, Ant-Dir, Hopper-Vel, and Walker-Vel. We interpolate the action representations between two tasks using Equation 1. The coefficient $\beta$ is searched to better fit the target task. An interpolation example in HalfCheetah-Vel is demonstrated in Figure 3. We select the tasks of running at 1 m/s and 2 m/s to be interpolated and get three interpolated tasks: run at 1.2 m/s, 1.5 m/s, 1.7 m/s. We perform evaluation on each task for a trajectory, and visualize them. In each task, we make the agent start from the same point (not plotted in the figure) and terminate at 100 time steps.

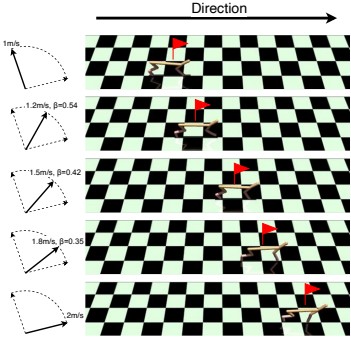

Figure 3: **Interpolated tasks.** The top and bottom rows are in-distribution. The middle rows show the interpolated tasks.

Only part of the whole scene is visualized in the figure and we mark the terminals with red flags. We find that through task interpolation, the agent manages to accomplish these interpolated tasks without any training. We leave more detailed interpolation results in all environments as well as some extrapolation trials to Appendix B.4.

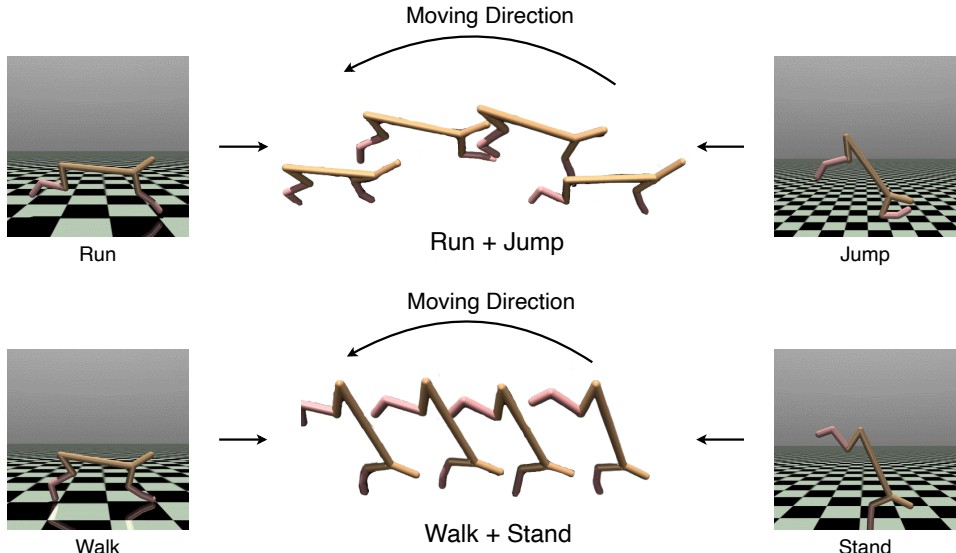

Figure 4: **Visualization of task compositions.** Two stop-motion animations of the proposed composition tasks are demonstrated. Animated results are shown in the project page: `https://sites.google.com/view/emergent-action-representation/`.

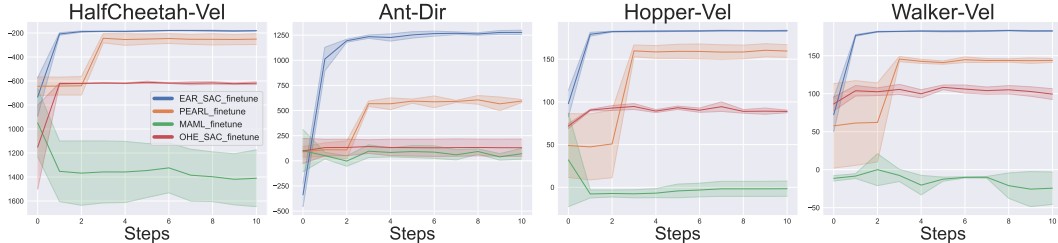

Figure 5: **Adaptation results.** The x-axis represents the adaptation steps and the y-axis represents the average reward of all the adaptation tasks. We fix the adaptation task set for all algorithms in the same environment.

**Inter-action composition.** Inter-action composition is conducted in HalfCheetah-Run-Jump, in which we merge two uni-modal task distributions. Taking HalfCheetah-BackVel and HalfCheetah-BackJump as an example, we find that the agent has learned four motions: walking backward, running backward, standing with its front leg, and jumping with its front leg. We select walk-standing as a composition task pair and run-jumping as the other and compose them with Equation 1. These two generated representations are evaluated in 1000 time steps and part of their evaluation trajectories are visualized in Figure 4. We find that the walk-standing representation enables the agent to walk backward with a standing posture despite some pauses to keep itself balance, while the run-jumping representation helps the agent to jump when running backward after some trials. These empirical results indicate that the LTEs can be used as action representations by composing them for different new tasks.

## 4.4 TASK ADAPTATION WITH ACTION REPRESENTATIONS

In this section, we assess how well an agent can adapt to new tasks by only updating the LTEs. We compare the agent's adaptation ability with the meta RL baselines (MAML,PEARL) and the ablation baseline (OHE-SAC). The results in Figure 5 demonstrate that EAR can adapt to new tasks and achieve to the converged performance in no more than three epochs, outperforming the baseline meta RL methods. We note that, with the help of the LTE space, we can adapt to new tasks using zero-th order optimization method.

## 4.5 VISUALIZATION OF STATE AND ACTION REPRESENTATIONS

In this section, we further analyze and visualize the sensory-action representations based on the HalfCheetah environment.

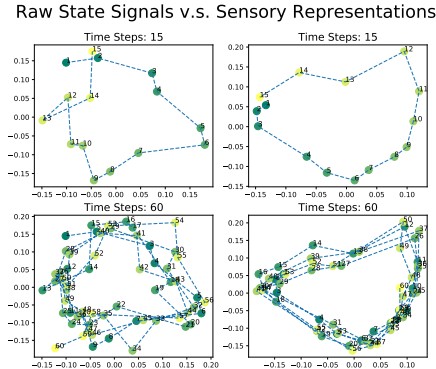

Action Representation Space

Figure 6: **Comparison between raw state signals (left) and sensory representations (right) in HalfCheetah-Vel.** To make the period of running motion easier to observe, we reduce the time interval between adjacent time steps. For now a period of the motion is 15 steps, while in the original setting it is only 6 steps.

Figure 7: **Visualization of the action representation space in HalfCheetah-Vel.** Each grid on the unit sphere represents the terminal of an action representation and is colored based on how the agent performs when the policy is conditioned on the action representation. The redder the grid is, the faster the agent runs.

**The sensory representations.** In HalfCheetah-Vel, the halfcheetah agent is encouraged to run at a specific velocity, thus making its motion periodical. Therefore, the ideal raw signals and the LSEs in the trajectory should be periodical as well. After conducting Principal Components Analysis (PCA) (Abdi & Williams, 2010) to reduce the dimension of all the raw signals and the sensory representations collected in a trajectory, we visualize them for different time steps in Figure 6. We find that the raw state signals appear to be only roughly periodical due to the noise in the states. However, the LSEs show stronger periodicity than the raw signals, indicating that the sensory representations can reduce the influence of the noise in raw signals, thus helping the agent better understand and react to the environment.

**The action representations.** To better understand the intriguing fact that the agent can adapt to new tasks using the emergent action representations when only sparse training tasks are provided, we plot the LTE space in Figure 7. We find that the LTEs on the hypersphere automatically form a continuous space, constructing the basis of the composition of the representations in it. This also explains why the interpolation and composition of the LTEs result in meaningful new behaviors: when interpolated and normalized, the new LTE would still lie on this hypersphere, leading to new and meaningful action sequences.

## 4.6 ABLATION STUDY

In this section, we ablate the two constraints mentioned in Section 3.1 as well as the overall representation-based architecture of our method to better understand what makes it work in the proposed structure. We find that the random noise injection improves the stability of the multi-task training procedure, while the unit sphere regularization improves the quality of the representation space and thus better supports task interpolation and composition. Further, we compare EAR with a different policy network, which takes in a raw state vector and a one-hot task embedding rather than the LSE and LTE, in the multi-task training and task adaptation settings. We find that the one-hot embedded policy performs unsatisfactorily in all environments and fails completely in the Ant-Dir environment, echoing with the idea that simple emergent action representations are meaningful and effective in task learning and generalization. A complete version of our ablation study which contains a more detailed analysis is in Appendix B.7.

## 5 RELATED WORK

**Representation learning in reinforcement learning.** Representation learning has been widely applied in RL to generate representations of sensory information (Laskin et al., 2020; Chandak et al., 2019; Yarats et al., 2021), policies (Edwards et al., 2019), dynamics (Watter et al., 2015).

In recent years, action representation has attracted more attention. In previous works (Dulac-Arnold et al., 2015), the action embeddings have been proposed to compress discrete actions to a continuous space based on prior knowledge. Recently, many researchers focus on mapping between the original continuous action space and a continuous manifold to facilitate and accelerate policy learning (Allshire et al., 2021) or simplify teleoperation (Losey et al., 2020). Moreover, such action representations are extended to high-dimensional real-world robotics control problems (Jeon et al., 2020). In this area, a closely related work is Chandak et al. (2019), which enables a policy to output a latent action representation with reinforcement learning and then uses supervised learning to construct a mapping from the representation space to the discrete action space. However, in our work, we discover emergent self-organized sensory-action representations from multi-task policy training without additional supervised learning, which can be further leveraged to generalize to new tasks.

**Multi-task reinforcement learning.** Multi-task RL trains an agent to learn multiple tasks simultaneously with a single policy. A direct idea is to share parameters and learn a joint policy with multiple objectives (Wilson et al., 2007; Pinto & Gupta, 2017), but different tasks may conflict with each other. Some works tackle this problem by reducing gradient conflicts (Yu et al., 2020a), designing loss weighting (Hessel et al., 2019; Kendall et al., 2018), or leveraging regularization (Duong et al., 2015). Modular network has also been proposed to construct the policy network with combination of sub-networks (Heess et al., 2016; Yang et al., 2020). In our work, multiple tasks share the state encoder and action decoder; the action decoder is conditioned on LTE to reduce task conflicts. Emergent representations are learned for continuous actions during multi-task training.

**Unsupervised skill discovery.** Unsupervised skill discovery employs competence-based methods to encourage exploration and help agents learn skills. These skills can be used in adapting to new tasks. The unsupervised methods maximize the mutual information between states and skills (Eysenbach et al., 2018; Liu & Abbeel, 2021; Sharma et al., 2019; Xu et al., 2020; Laskin et al., 2022), while our method learns a geometrically and semantically meaningful representation in a multi-task setting. We believe the proposed method can be complementary to the unsupervised paradigm.

**Meta reinforcement learning.** The adaptation part of this work shares a similar goal as meta reinforcement learning (meta RL) (Finn et al., 2017; Xu et al., 2018a; Rothfuss et al., 2018; Wang et al., 2016; Levine et al., 2020). Meta RL executes meta-learning (Schmidhuber, 1987; Thrun & Pratt, 2012) algorithms under the framework of reinforcement learning, which aims to train an agent to quickly adapt to new tasks. Many meta RL methods are gradient-based, performing gradient descent on the policy (Finn et al., 2017; Xu et al., 2018a; Rothfuss et al., 2018), the hyperparameters (Xu et al., 2018b), or loss functions (Sung et al., 2017; Houthooft et al., 2018) based on the collected experience. Other meta RL algorithms are context-based (Wang et al., 2016; Rakelly et al., 2019). They embed context information into a latent variable and condition the meta-learning policy on such variables. Besides, recently offline meta RL algorithms (Levine et al., 2020; Mitchell et al., 2021) have also been widely explored, which enables the agent to leverage offline data to perform meta-training. However, we exploit the emergent high-level representations to abstract low-level actions for adaptation rather than introduce complicated new techniques.

## 6 CONCLUSION

In this paper, we present our finding that the task embeddings in a multi-task policy network can automatically form a representation space of low-level motor signals, where the semantic structure of actions is explicit. The emergent action representations abstract information of a sequence of actions and can serve as a high-level interface to instruct an agent to perform motor skills. Specifically, we find the action representations can be interpolated, composed, and optimized to generate novel action sequences for unseen tasks. Along with this work, a promising direction is to learn the action representation via self-supervised learning. Another intriguing direction is to learn hierarchical action representations that may capture different levels of semantics and geometric information in motor signals, thus facilitating future applications such as hierarchical planning (LeCun, 2022).

ACKNOWLEDGEMENT

Yubei would like to thank Yann LeCun and Rob Fergus for sharing their visionary thoughts on action representation learning. We also thank Shaoxiong Yao for participating in the early discussion of this work.

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

## A  DETAILED DESCRIPTION OF SAC

SAC is a stable off-policy actor-critic algorithm based on the maximum entropy reinforcement learning framework, in which the actor maximizes both the returns and the entropy. In SAC, there are three types of parameters to update during optimization: the policy parameter $\omega$, the soft Q-function parameter $\theta$ and a learnable temperature $\alpha$. The objectives are:

$$J(\alpha) = \mathbb{E}_{\mathbf{a}_t \sim \pi_\omega} \left[ -\alpha \log \pi_\omega \left( \mathbf{a}_t \mid \mathbf{s}_t \right) - \alpha \overline{\mathcal{H}} \right] \tag{2}$$

$$J_\pi(\omega) = \mathbb{E}_{\mathbf{s}_t \sim \mathcal{D}} \left[ \mathbb{E}_{\mathbf{a}_t \sim \pi_\omega} \left[ \alpha \log \pi_\omega \left( \mathbf{a}_t \mid \mathbf{s}_t \right) - Q_\theta \left( \mathbf{s}_t, \mathbf{a}_t \right) \right] \right] \tag{3}$$

$$J_Q(\theta) = \mathbb{E}_{(\mathbf{s}_t, \mathbf{a}_t) \sim \mathcal{D}} \left[ \frac{1}{2} \left( Q_\theta \left( \mathbf{s}_t, \mathbf{a}_t \right) - \left( r \left( \mathbf{s}_t, \mathbf{a}_t \right) + \gamma \mathbb{E}_{\mathbf{s}_{t+1} \sim p} \left[ V_{\bar{\theta}} \left( \mathbf{s}_{t+1} \right) \right] \right) \right)^2 \right] \tag{4}$$

where $\overline{\mathcal{H}}$ in Equation 2 is a pre-defined minimum expected entropy. The value function in Equation 4 is parameterized by the weights $V_{\bar{\theta}}$ in the target Q-network and can be defined by:

$$V \left( \mathbf{s}_t \right) = \mathbb{E}_{\mathbf{a}_t \sim \pi} \left[ Q \left( \mathbf{s}_t, \mathbf{a}_t \right) - \alpha \log \pi \left( \mathbf{a}_t \mid \mathbf{s}_t \right) \right] \tag{5}$$

## B  EXPERIMENTAL DETAILS

### B.1  ENVIRONMENTS

The detailed descriptions of the reinforcement learning benchmarks in our experiments are listed as follows:

- **HalfCheetah-Vel (Uni-modal):** In this environment, we train the halfcheetah agent to run at a target velocity. The training task set contains 10 velocities. The target velocities during training range from 1 m/s to 10 m/s. For every 1 m/s, we set a training task. The adaptation task set contains 3 velocities that are uniformly sampled from [1,10]. The agent is penalized with the $l_1$ error between its velocity and the target velocity.

- **Ant-Dir (Uni-modal):** In this environment, we train the Ant agent to run in a target direction. The training task set contains 24 directions which are consecutive integers from 1 to 10. The adaptation task set contains 5 integer directions which are uniformly sampled from [0,360) degrees. The agent is rewarded with the velocity along the target direction and penalized with the velocity along the direction perpendicular to our target.

- **Hopper/Walker-Vel (Uni-modal):** Similar to HalfCheetah-Vel, Hopper/Walker-Vel contains 10 velocities in training task set and 3 in adaptation task set. The target velocities when training range from 0.2 m/s to 2 m/s and every 0.2 m/s we set a training task. The Hopper/Walker agent is penalized with the $l_1$ error between its velocity and the target velocity.

- **HalfCheetah-Run-Jump (Multi-modal):** There are two different uni-modal task distributions in this environment, respectively HalfCheetah-BackVel and HalfCheetah-BackJump. HalfCheetah-BackVel trains the halfcheetah agent to run backward at a target velocity, and the agent is penalized with the $l_1$ error between its velocity and the target velocity. HalfCheetah-BackJump trains the agent to raise its hind leg to jump, and the agent is rewarded with the height of its hind leg. HalfCheetah-Run-Jump contains 7 velocities for running backward (from 1 m/s to 7 m/s and every 1 m/s a training task is set) and 3 tasks for jumping (3 different weights for the leg height in the total reward function) in the training task set. In our implementation, the jumping task with smallest weight finally turns out to be a motion of "standing" instead of jumping.

### B.2  BASELINES

The detailed descriptions of the multi-task RL and meta RL baselines are listed as follows:

- **MAML.** MAML (Finn et al., 2017) is a gradient-based meta-RL algorithm based on Trust Region Policy Optimization (Schulman et al., 2015). It aims to learn easily adaptable model parameters for tasks in a specific task distribution. MAML conducts explicit training on model parameters so that a small number of training data and gradient steps are required to adapt to a new task. In our implementation, we modify the range and density of task sampling, which will be discussed in the following section.

- **PEARL.** PEARL (Rakelly et al., 2019) is a context-based off-policy meta-RL algorithm based on Soft Actor Critic (Haarnoja et al., 2018) and variational inference method (Kingma & Welling, 2013). PEARL tackles the problem of limited sample efficiency in meta-RL domain. PEARL integrates off-policy RL algorithms with a latent task variable which infers how to adapt to a new task with small amount of history contexts.

- **Multi-head multi-task SAC (MHMT-SAC).** MHMT-SAC is a multi-task RL architecture, in which each training task has an independent head input, and the body of the policy network is shared among all the tasks. We compare our method with MHMT-SAC to evaluate the performance of an agent with emergent action representation in multi-task training.

- **One-hot embedding SAC (OHE-SAC).** The sensory-action representation is introduced when training the agent to learn multiple tasks in the training set. In OHE-SAC, We omit the representation layer to make the policy network a common multi-task RL structure, which concatenates a one-hot embedding and the raw state vector as the input of the policy network. Through comparison between the final performance of the two methods, OHE-SAC helps identify the effect of action representations in training and adaptation.

### B.3 Multi-task training

#### B.3.1 Network Architecture

In this section, we discuss the choice of network architecture. In our method, we use MLP as the network structure of the encoders and decoders. Here we replace the first layer of MLP with an RNN layer to get the RNN encoder and decoder. Our MLP-based network is compared with the RNN-based network. A comparison in the HalfCheetah-Vel environment of the two networks is shown in Figure 8. We find that if we simply make such changes with a certain amount of tuning, our simple MLP-based network achieves much higher performance than an RNN-based one.

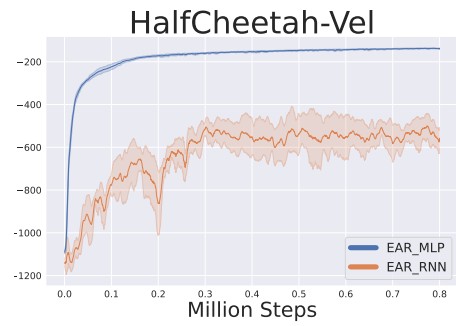

Figure 8: Comparison of MLP and RNN structure.

#### B.3.2 Hyperparameters

In this section, we provide detailed settings of our methods. We set up the hyperparameters, as shown in Table 2, for the environments and algorithms in the Mujoco locomotion benchmarks.

#### B.3.3 Task sampling density

In the experiments, we fix the range of task sampling to be for different algorithms in the same environment. We find that in HalfCheetah-Vel environment, multi-task RL methods get different final rewards for different tasks (in Figure 9). The reason is that when the agent is trained well enough, the penalty of velocity in the reward will become slight, thus making the control cost of the agent dominant in the reward function. Therefore, it is natural that the faster the agent runs, the larger cost it should pay, leading to a relatively low reward. Thus to make our comparison valid and persuasive, the range of task sampling should be fixed. The detailed settings of the implementation in our paper are demonstrated in Table 3. Similarly, the adaptation tasks are also shared among all the algorithms in the same environment in Section 4.4.

### B.4 Intra-action interpolation

**Additional details for the "Intra-action interpolation".** We have performed task interpolation between adjacent tasks in HalfCheetah-Vel in Section 4.3, where we can interpolate the LTEs of running at 1m/s and 2m/s to control the agent to run at 1.2, 1.5, 1.7m/s. Some visualized results are provided in the text, and here we demonstrate a quantitative analysis based on multiple seeds for

| Hyperparameters | HalfCheetah-Vel | Hopper/Walker-Vel | Ant-Dir |
|---|---|---|---|
| max episode frames | 200 | 200 | 200 |
| sensory encoder | 2-layer MLP | 2-layer MLP | 2-layer MLP |
| action decoder | 4-layer MLP | 4-layer MLP | 4-layer MLP |
| task encoder | 1-layer FC | 1-layer FC | 1-layer FC |
| Q-network | 4-layer MLP | 4-layer MLP | 4-layer MLP |
| LSE shape | 16 | 16 | 16 |
| LTE shape | 3 | 3 | 3 |
| optimizer | Adam | Adam | Adam |
| learning rate | 3 | 3 | 3 |
| discount | 0.99 | 0.99 | 0.99 |
| batch size | 1280 | 1280 | 1920 |
| replay buffer | 1e6 | 1e6 | 1.2e6 |
| pretrain epochs | 20 | 20 | 20 |
| training epochs | 4k | 5k | 10k |
| optimization times | 200 | 200 | 200 |
| evaluation episodes | 3 | 3 | 3 |

Table 2: Hyperparameters of EAR in four benchmarks

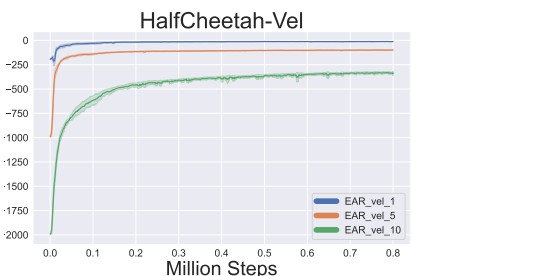
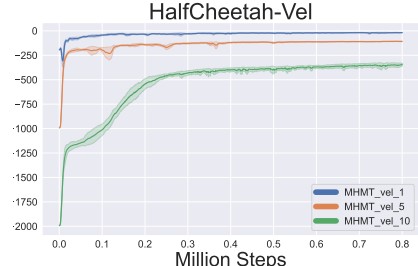

Figure 9: Training curves of tasks with different target velocities with multi-task RL method.

the statistical reliability of our claims. During the experiments, we find that the agent may learn to move forward by hand-standing instead of running when faced with the 1m/s task in some seeds. It makes the postures of 1m/s and 2m/s rather different and hard to achieve interpolation. Due to such a phenomenon, we change the pre-trained tasks to 2m/s and 3m/s, and the interpolated tasks to 2.2m/s, 2.5m/s and 2.7m/s to fairly assess the stability of task interpolation on multiple seeds. The coefficients $\beta$ of the task interpolation on 3 seeds are demonstrated in Table 4. We can find that as the coefficient of interpolation decreases, the cheetah tends to run faster, which indicates that the LTEs between the embeddings of running at 2m/s and 3m/s show continuity in the LTE Space.

**Mid-task interpolation.** Meanwhile, to fully evaluate the performance of task interpolation, we conduct another experiment on all environments. We aim to adjust the interpolation coefficients to achieve "mid-task interpolation" between pairs of tasks, where a "mid-task" has the average modality of the two base tasks. For instance, in the first seed of the experiment aforementioned, we find that when $\beta = 0.42$, the agent can run forward at 2.5m/s, which is a mid-task of running at 2m/s and 3m/s. The results of mid-task interpolation in 3 seeds are shown in Table 11,12,13,14.

**1:1 task interpolation.** In the previous parts, we first determine the desired interpolated tasks and find a suitable coefficient $\beta$ for the task. In this part, from another angle, we fix the interpolation co-efficient and observe the corresponding tasks. We perform 1:1 task interpolation in all environments, which means we fix the coefficients $\beta$ to always be 0.5. The results are shown in Table 15,16,17,18. We summarize the results of the interpolation aforementioned in Table 5.

**Task extrapolation trials.** Besides interpolation, we also perform extrapolation between tasks, which means we make $\beta > 1$ or $\beta < 0$, in HalfCheetah-Vel and Ant-Dir. We demonstrate some of the results in Table 6. We find that, in HalfCheetah-Vel, the extrapolated behavior can also achieve the target speed as long as the speed is in the distribution of the training tasks. The exception happens

| Method | HalfCheetah-Vel | Hopper/Walker-Vel | Ant-Dir |
|---|---|---|---|
| **EAR-SAC (Ours)** | **10 tasks in [0,10]** | **10 tasks in [0,2]** | **24 tasks in [0,360)** |
| MAML | 100 tasks in [0,10] | 40 tasks in [0,2] | 40 tasks in [0,360) |
| PEARL | 100 tasks in [0,10] | 20 tasks in [0,2] | 100 tasks in [0,360) |
| MTMH-SAC | 10 tasks in [0,10] | 10 tasks in [0,2] | 24 tasks in [0,360) |
| OHE-SAC | 10 tasks in [0,10] | 10 tasks in [0,2] | 24 tasks in [0,360) |

Table 3: Density of task sampling in the training procedure.

| Task | $\beta$ | Evaluation | Task | $\beta$ | Evaluation | Task | $\beta$ | Evaluation |
|---|---|---|---|---|---|---|---|---|
| Vel-2.0 | 1 | 2.01m/s | Vel-2.0 | 1 | 2.00m/s | Vel-2.0 | 1 | 2.00m/s |
| **Vel-2.2** | 0.67 | 2.20m/s | **Vel-2.2** | 0.64 | 2.20m/s | **Vel-2.2** | 0.69 | 2.20m/s |
| **Vel-2.5** | 0.42 | 2.50m/s | **Vel-2.5** | 0.42 | 2.50m/s | **Vel-2.5** | 0.44 | 2.51m/s |
| **Vel-2.7** | 0.25 | 2.70m/s | **Vel-2.7** | 0.30 | 2.69m/s | **Vel-2.7** | 0.29 | 2.70m/s |
| Vel-3.0 | 0 | 2.98m/s | Vel-3.0 | 0 | 2.99m/s | Vel-3.0 | 0 | 3.00m/s |

Table 4: Coefficients and evaluation results of task interpolation in HalfCheetah-Vel.

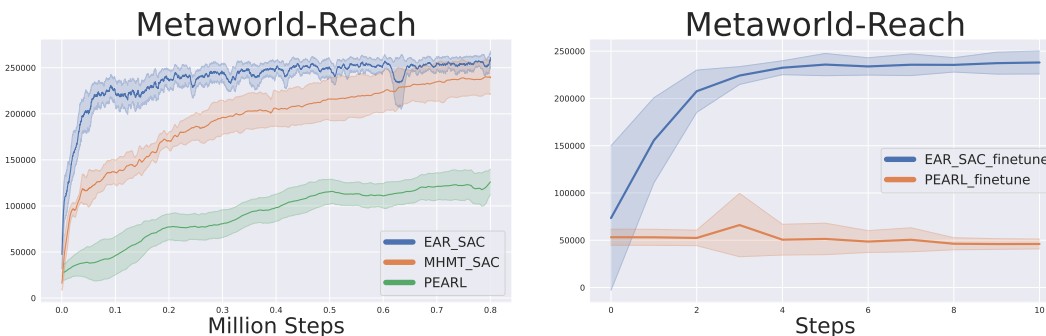

Figure 10: Training performance of our method and baselines in Metaworld-Reach.

Figure 11: Adaptation results of our method and PEARL in Metaworld-Reach.

when the target speed is out of the distribution(e.g. extrapolate Vel-1.0 and Vel-2.0 with $\beta > 1$ or Vel-9.0 and Vel-10.0 with $\beta < 0$) where the agent can only perform extrapolation within a relatively small range of $\beta$. In the higher-dimensional environment Ant-Dir, extrapolation still works in most cases but has a higher failure rate.

### B.5 INTER-ACTION COMPOSITION

We have performed task composition in the HalfCheetah-Run-Jump environment. Two stop motion animations are shown in Section 4.3 and we provide more quantitative results here. The coefficients $\beta$ of the task composition in multiple seeds are demonstrated in Table 10. Composing walking and standing succeeds in all three seeds while composing running and jumping only succeeds in two seeds. In the seed where the run-jumping task fails, we discover a new motion called move-jumping, in which the cheetah agent jumps backward with its hind leg raised.

### B.6 APPLICATION IN MANIPULATION

In previous sections, we mainly tackle the classic and widely studied locomotion problems in reinforcement learning. In this section, we apply our method to the domain of manipulation to examine the effectiveness of the proposed method in other domains. We propose another environment Metaworld-Reach based on the metaworld (Yu et al., 2020b), in which a robot arm is trained to reach different points. We train the agent with multiple reach goals under the proposed multi-task setting and perform fast adaptation with the learned policy. The training and adaptation curves are

| Success Rate | HalfCheetah-Vel | Hopper-Vel | Walker-Vel | Ant-Dir |
|---|---|---|---|---|
| Mid-task interpolation | 26/27 | 27/27 | 27/27 | 70/72 |
| 1:1 task interpolation | 25/27 | 27/27 | 25/27 | 66/72 |

Table 5: Success rate of mid-task interpolation and 1:1 task interpolation on all environments. The definition of success in mid-task interpolation is whether a suitable $\beta \in (0.1, 0.9)$ can be found, while in 1:1 task interpolation is whether the interpolated task lies between the two base tasks.

| Tasks | $\beta$ | Evaluation | Tasks | $\beta$ | Evaluation | Tasks | $\beta$ | Evaluation |
|---|---|---|---|---|---|---|---|---|
| Vel-1.0 & Vel-2.0 | 1.8 | 1.04 | Vel-4.0 & Vel-5.0 | 1.8 | 3.03 | Vel-9.0 & Vel-10.0 | 1.8 | 8.12 |
| | 1.5 | 1.01 | | 1.5 | 3.50 | | 1.5 | 8.45 |
| | 1.2 | 0.99 | | 1.2 | 3.84 | | 1.2 | 8.76 |
| | -0.2 | 2.07 | | -0.2 | 5.21 | | -0.2 | 10.10 |
| | -0.5 | 2.19 | | -0.5 | 5.53 | | -0.5 | 10.29 |
| | -0.8 | 2.35 | | -0.8 | 5.87 | | -0.8 | 10.19 |

| Tasks | $\beta$ | Evaluation | Tasks | $\beta$ | Evaluation | Tasks | $\beta$ | Evaluation |
|---|---|---|---|---|---|---|---|---|
| Dir-0 & Dir-15 | 1.8 | 288.73 | Dir-120 & Dir-135 | 1.8 | 107.03 | Dir-270 & Dir-285 | 1.8 | 255.95 |
| | 1.5 | 308.72 | | 1.5 | 111.38 | | 1.5 | 259.95 |
| | 1.2 | 342.61 | | 1.2 | 116.81 | | 1.2 | 268.35 |
| | -0.2 | 9.46 | | -0.2 | 135.53 | | -0.2 | 288.98 |
| | -0.5 | 355.72 | | -0.5 | 141.67 | | -0.5 | 288.36 |
| | -0.8 | 49.29 | | -0.8 | 145.02 | | -0.8 | 294.08 |

Table 6: Coefficients and evaluation results of task extrapolation in HalfCheetah-Vel and Ant-Dir in seed 3.

demonstrated respectively in Figure 10,11. We find that our method can still find suitable latent task embeddings in the action representation space for adaptation tasks in less than 5 steps.

### B.7 COMPLETE VERSION OF ABLATION STUDY

In this section, we ablate the two constraints and the overall representation-based architecture of our method to better understand what makes it work in the proposed structure.

**Random Noise Injection.** We inject random noise into the LTEs during the training procedure to enhance the smoothness of the space. To assess the effect of this injected noise, we compare it with the case without the injected noise in the high-dimensional environment Ant-Dir. The training curves are shown in Figure 12. We observe that, when injected with random noise, our method achieves comparable training performance to the one without noise. Then we evaluate the policy we learn. For each task in each seed, we evaluate the trained policy for multiple trajectories. We find that without the injected noise in training, the policy will be sensitive to perturbation in the system during evaluation. An example of such a phenomenon is shown in Table 7. In the task of "direction-240" in Ant-Dir, we evaluate the policy for 5 trajectories and find that the second and third trials terminate before the maximal steps (200 steps), which means they fail halfway. We count the number of such failures respectively and summarize them in Table 8. The policy trained without injected noise fails to perform over 20% of tasks stably, while the one with noise avoids failure.

| No. | Terminate steps |
|---|---|
| Trajectory 1 | 200 |
| Trajectory 2 | 102 |
| Trajectory 3 | 144 |
| Trajectory 4 | 200 |
| Trajectory 5 | 200 |

Table 7: An example: Direction-240 task in Ant-Dir in seed 5.

| Failure Rate | Number of failures |
|---|---|
| Without noise | 16 in 72 tasks |
| With noise | **0 in 72 tasks** |

Table 8: Failure rates among all training tasks with/without injected noise. There are 24 tasks in Ant-Dir for each seed and 3 seeds in total.

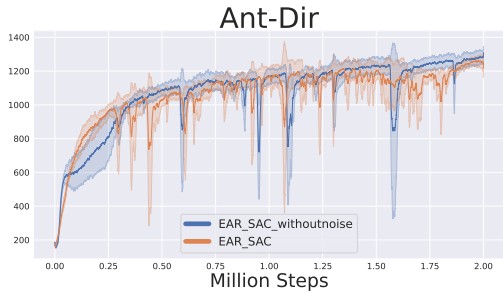
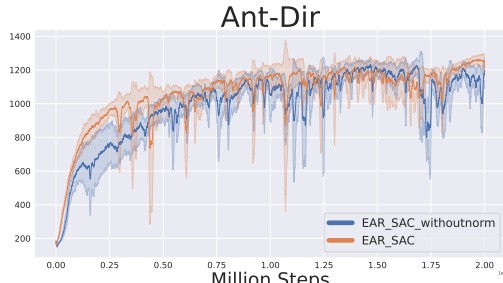

Figure 12: The comparison of training curves of the policy trained with/without the injected random noise.

Figure 13: The comparison of training curves of the policy with normalized/none-normalized LTEs and LSEs.

| Success Rate | Mid-task interpolation | 1:1 task interpolation |
|---|---|---|
| Without normalization | 64/72 | 59/72 |
| With normalization | **70/72** | **66/72** |

Table 9: Success rates of mid-task and 1:1 task interpolation with/without normalization.

**Unit sphere regularization.** We also apply regularization on LTEs by normalizing them on the surface of a unit sphere. We ablate this regularization by removing the normalization functions for LSEs and LTEs and directly concatenating them together as the input of the action decoder. The training curves are demonstrated in Figure 13. We find that our method achieves higher sample efficiency and rewards than the one without normalization. Then we compare their performance in downstream tasks. We repeat the interpolation experiments with the new policy without normalization and calculate the success rates in Table 9. With the normalization, the success rates of interpolation experiments increase, which indicates that normalization improves the quality of the representation space and thus better supports interpolation. Furthermore, we also try to perform task adaptation with the new policy. An example is demonstrated in Figure 14. Results show that the new policy can still adapt to some tasks while requiring more training steps.

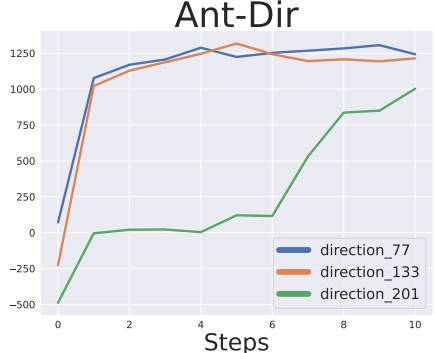

Figure 14: An example: three random adaptation tasks in seed 1. Two achieve comparable results with our method while the remaining one requires much more steps.

**Representation-based architecture.** Readers might be curious about whether the learned, manipulable LTEs are better than a naive one-hot embedding. In this part, we compare EAR with a different policy network that takes in a raw state vector and a one-hot task embedding rather than the LSE and LTE. Specifically, in multi-task training, we concatenate the raw state vector with the one-hot embedding as the input of the policy network to form a multi-task RL architecture. From the results in Figure 2 and Table 1, we find that the one-hot embedded policy performs unsatisfactorily in all environments and fails completely in the Ant-Dir environment. We attribute this to the fact that, without the sensory-action representation, the agent fails to understand the underlying relationships among multiple tasks, especially in those environments with high-dimensional observations and a large training task set. We also compare both methods in the task adaptation setting in Figure 5. When optimizing the one-hot embedding to adapt to new tasks, the agent can barely reach a reasonable performance in any of the new tasks. These facts echo with the idea that simple emergent action representations are meaningful and effective in task learning and generalization.

| Task | $\beta$ |
|---|---|
| Move-jumping | 0.41 |
| Walk-standing | 0.42 |

| Task | $\beta$ |
|---|---|
| Run-jumping | 0.52 |
| Walk-standing | 0.30 |

| Task | $\beta$ |
|---|---|
| Run-jumping | 0.27 |
| Walk-standing | 0.45 |

Table 10: Coefficients of task composition in HalfCheetah-Run-Jump.

| Target | 1.5 | 2.5 | 3.5 | 4.5 | 5.5 | 6.5 | 7.5 | 8.5 | 9.5 |
|---|---|---|---|---|---|---|---|---|---|
| $\beta$ | 0.47 | 0.44 | 0.38 | 0.44 | 0.46 | 0.46 | 0.47 | 0.44 | 0.46 |
| Evaluation | 1.60 | 2.51 | 3.51 | 4.50 | 5.50 | 6.50 | 7.51 | 8.50 | 9.51 |
| Target | 1.5 | 2.5 | 3.5 | 4.5 | 5.5 | 6.5 | 7.5 | 8.5 | 9.5 |
| $\beta$ | 0.42 | 0.42 | 0.38 | 0.39 | 0.43 | 0.45 | 0.46 | 0.51 | 0.48 |
| Evaluation | 1.51 | 2.50 | 3.50 | 4.50 | 5.50 | 6.50 | 7.50 | 8.50 | 9.50 |
| Target | 1.5 | 2.5 | 3.5 | 4.5 | 5.5 | 6.5 | 7.5 | 8.5 | 9.5 |
| $\beta$ | 0.37 | 0.42 | 0.45 | 0.43 | 0.49 | 0.52 | 0.51 | 0.53 | 0.41 |
| Evaluation | 1.49 | 2.50 | 3.50 | 4.50 | 5.50 | 6.50 | 7.50 | 8.49 | 9.51 |

Table 11: Mid-task interpolation in HalfCheetah-Vel.

| Target | 0.3 | 0.5 | 0.7 | 0.9 | 1.1 | 1.3 | 1.5 | 1.7 | 1.9 |
|---|---|---|---|---|---|---|---|---|---|
| $\beta$ | 0.59 | 0.50 | 0.47 | 0.39 | 0.51 | 0.64 | 0.55 | 0.64 | 0.60 |
| Evaluation | 0.30 | 0.50 | 0.70 | 0.90 | 1.11 | 1.30 | 1.50 | 1.70 | 1.90 |
| Target | 0.3 | 0.5 | 0.7 | 0.9 | 1.1 | 1.3 | 1.5 | 1.7 | 1.9 |
| $\beta$ | 0.59 | 0.44 | 0.45 | 0.54 | 0.63 | 0.54 | 0.53 | 0.60 | 0.61 |
| Evaluation | 0.30 | 0.51 | 0.70 | 0.90 | 1.09 | 1.30 | 1.50 | 1.70 | 1.90 |
| Target | 0.3 | 0.5 | 0.7 | 0.9 | 1.1 | 1.3 | 1.5 | 1.7 | 1.9 |
| $\beta$ | 0.51 | 0.52 | 0.52 | 0.58 | 0.48 | 0.72 | 0.63 | 0.58 | 0.04 |
| Evaluation | 0.30 | 0.50 | 0.70 | 0.91 | 1.10 | 1.30 | 1.50 | 1.70 | 1.90 |

Table 12: Mid-task interpolation in Hopper-Vel.

| Target | 0.3 | 0.5 | 0.7 | 0.9 | 1.1 | 1.3 | 1.5 | 1.7 | 1.9 |
|---|---|---|---|---|---|---|---|---|---|
| $\beta$ | 0.54 | 0.36 | 0.49 | 0.38 | 0.45 | 0.44 | 0.66 | 0.82 | 0.77 |
| Evaluation | 0.30 | 0.50 | 0.70 | 0.90 | 1.10 | 1.30 | 1.50 | 1.70 | 1.90 |
| Target | 0.3 | 0.5 | 0.7 | 0.9 | 1.1 | 1.3 | 1.5 | 1.7 | 1.9 |
| $\beta$ | 0.36 | 0.23 | 0.37 | 0.44 | 0.41 | 0.45 | 0.48 | 0.58 | 0.74 |
| Evaluation | 0.30 | 0.50 | 0.70 | 0.90 | 1.10 | 1.30 | 1.51 | 1.70 | 1.90 |
| Target | 0.3 | 0.5 | 0.7 | 0.9 | 1.1 | 1.3 | 1.5 | 1.7 | 1.9 |
| $\beta$ | 0.50 | 0.39 | 0.46 | 0.51 | 0.43 | 0.52 | 0.51 | 0.76 | 0.62 |
| Evaluation | 0.30 | 0.50 | 0.70 | 0.90 | 1.10 | 1.30 | 1.50 | 1.70 | 1.90 |

Table 13: Mid-task interpolation in Walker-Vel.

| Target | 7.5 | 22.5 | 37.5 | 52.5 | 67.5 | 82.5 | 97.5 | 112.5 |
|---|---|---|---|---|---|---|---|---|
| $\beta$ | 0.45 | 0.82 | 0.46 | 0.46 | 0.49 | 0.50 | 0.34 | 0.40 |
| Evaluation | 7.48 | 22.51 | 37.81 | 52.57 | 67.56 | 82.50 | 97.62 | 112.29 |
| Target | 127.5 | 142.5 | 157.5 | 172.5 | 187.5 | 202.5 | 217.5 | 232.5 |
| $\beta$ | 0.49 | 0.68 | 0.42 | 0.99 | 0.73 | 0.58 | 0.36 | 0.70 |
| Evaluation | 127.58 | 142.47 | 157.47 | 163.43 | 187.48 | 202.53 | 217.39 | 232.52 |
| Target | 247.5 | 262.5 | 277.5 | 292.5 | 307.5 | 322.5 | 337.5 | 352.5 |
| $\beta$ | 0.26 | 0.67 | 0.58 | 0.31 | 0.36 | 0.37 | 0.57 | 0.47 |
| Evaluation | 247.50 | 262.48 | 277.50 | 292.61 | 307.56 | 322.49 | 337.55 | 352.42 |
| Target | 7.5 | 22.5 | 37.5 | 52.5 | 67.5 | 82.5 | 97.5 | 112.5 |
| $\beta$ | 0.76 | 0.56 | 0.37 | 0.20 | 0.59 | 0.31 | 0.40 | 0.40 |
| Evaluation | 7.27 | 22.54 | 37.51 | 52.32 | 67.54 | 82.58 | 95.52 | 112.52 |
| Target | 127.5 | 142.5 | 157.5 | 172.5 | 187.5 | 202.5 | 217.5 | 232.5 |
| $\beta$ | 0.34 | 0.22 | 0.52 | 0.48 | 0.41 | 0.47 | 0.38 | 0.27 |
| Evaluation | 127.69 | 142.80 | 157.56 | 172.39 | 187.50 | 202.50 | 217.44 | 232.61 |
| Target | 247.5 | 262.5 | 277.5 | 292.5 | 307.5 | 322.5 | 337.5 | 352.5 |
| $\beta$ | 0.35 | 0.56 | 0.40 | 0.35 | 0.86 | 0.52 | 0.66 | 0.48 |
| Evaluation | 247.50 | 262.33 | 277.49 | 292.42 | 307.37 | 322.55 | 337.32 | 352.82 |
| Target | 7.5 | 22.5 | 37.5 | 52.5 | 67.5 | 82.5 | 97.5 | 112.5 |
| $\beta$ | 0.75 | 0.50 | 0.53 | 0.29 | 0.46 | 0.43 | 0.39 | 0.99 |
| Evaluation | 7.58 | 22.48 | 37.44 | 52.44 | 67.63 | 82.53 | 97.61 | 112.62 |
| Target | 127.5 | 142.5 | 157.5 | 172.5 | 187.5 | 202.5 | 217.5 | 232.5 |
| $\beta$ | 0.55 | 0.47 | 0.34 | 0.36 | 0.30 | 0.21 | 0.27 | 0.32 |
| Evaluation | 127.51 | 142.44 | 157.37 | 172.31 | 187.43 | 202.68 | 217.59 | 232.48 |
| Target | 247.5 | 262.5 | 277.5 | 292.5 | 307.5 | 322.5 | 337.5 | 352.5 |
| $\beta$ | 0.26 | 0.30 | 0.30 | 0.38 | 0.15 | 0.55 | 0.47 | 0.41 |
| Evaluation | 247.44 | 262.54 | 277.45 | 292.84 | 307.19 | 322.44 | 337.74 | 352.33 |

Table 14: Mid-task interpolation in Ant-Dir.

| Target | 1.5 | 2.5 | 3.5 | 4.5 | 5.5 | 6.5 | 7.5 | 8.5 | 9.5 |
|---|---|---|---|---|---|---|---|---|---|
| Evaluation | 0.96 | 2.42 | 3.30 | 4.43 | 5.46 | 6.45 | 7.49 | 8.46 | 9.49 |
| Target | 1.5 | 2.5 | 3.5 | 4.5 | 5.5 | 6.5 | 7.5 | 8.5 | 9.5 |
| Evaluation | 1.23 | 2.38 | 3.35 | 4.40 | 5.41 | 6.46 | 7.47 | 8.44 | 9.02 |
| Target | 1.5 | 2.5 | 3.5 | 4.5 | 5.5 | 6.5 | 7.5 | 8.5 | 9.5 |
| Evaluation | 1.25 | 2.38 | 3.42 | 4.42 | 5.48 | 6.54 | 7.49 | 8.50 | 9.43 |

Table 15: 1:1 interpolation in HalfCheetah-Vel.

| Target | 0.3 | 0.5 | 0.7 | 0.9 | 1.1 | 1.3 | 1.5 | 1.7 | 1.9 |
|---|---|---|---|---|---|---|---|---|---|
| Evaluation | 0.31 | 0.51 | 0.68 | 0.91 | 1.08 | 1.36 | 1.54 | 1.72 | 1.96 |
| Target | 0.3 | 0.5 | 0.7 | 0.9 | 1.1 | 1.3 | 1.5 | 1.7 | 1.9 |
| Evaluation | 0.31 | 0.46 | 0.69 | 0.90 | 1.14 | 1.31 | 1.50 | 1.73 | 1.95 |
| Target | 0.3 | 0.5 | 0.7 | 0.9 | 1.1 | 1.3 | 1.5 | 1.7 | 1.9 |
| Evaluation | 0.38 | 0.51 | 0.69 | 0.89 | 1.15 | 1.34 | 1.51 | 1.73 | 1.90 |

Table 16: 1:1 interpolation in Hopper-Vel.

| Target | 0.3 | 0.5 | 0.7 | 0.9 | 1.1 | 1.3 | 1.5 | 1.7 | 1.9 |
|---|---|---|---|---|---|---|---|---|---|
| Evaluation | 0.32 | 0.46 | 0.69 | 0.87 | 1.09 | 1.29 | 1.82 | 1.83 | 1.97 |
| Target | 0.3 | 0.5 | 0.7 | 0.9 | 1.1 | 1.3 | 1.5 | 1.7 | 1.9 |
| Evaluation | 0.26 | 0.46 | 0.67 | 0.89 | 1.08 | 1.28 | 1.51 | 1.72 | 1.97 |
| Target | 0.3 | 0.5 | 0.7 | 0.9 | 1.1 | 1.3 | 1.5 | 1.7 | 1.9 |
| Evaluation | 0.29 | 0.47 | 0.69 | 0.90 | 1.09 | 1.30 | 1.50 | 1.79 | 1.95 |

Table 17: 1:1 interpolation in Walker-Vel.

| Target | 7.5 | 22.5 | 37.5 | 52.5 | 67.5 | 82.5 | 97.5 | 112.5 |
|---|---|---|---|---|---|---|---|---|
| Evaluation | 8.65 | 22.86 | 38.86 | 52.07 | 67.76 | 79.79 | 175.41 | 113.61 |
| Target | 127.5 | 142.5 | 157.5 | 172.5 | 187.5 | 202.5 | 217.5 | 232.5 |
| Evaluation | 126.45 | 142.40 | 154.05 | 169.70 | 185.50 | 198.52 | 214.90 | 231.39 |
| Target | 247.5 | 262.5 | 277.5 | 292.5 | 307.5 | 322.5 | 337.5 | 352.5 |
| Evaluation | 243.37 | 258.17 | 273.74 | 290.81 | 316.52 | 323.48 | 338.66 | 351.11 |
| Target | 7.5 | 22.5 | 37.5 | 52.5 | 67.5 | 82.5 | 97.5 | 112.5 |
| Evaluation | 47.26 | 25.14 | 39.65 | 26.37 | 60.58 | 81.91 | 96.71 | 114.74 |
| Target | 127.5 | 142.5 | 157.5 | 172.5 | 187.5 | 202.5 | 217.5 | 232.5 |
| Evaluation | 125.47 | 133.68 | 160.27 | 171.79 | 190.62 | 201.69 | 217.45 | 229.55 |
| Target | 247.5 | 262.5 | 277.5 | 292.5 | 307.5 | 322.5 | 337.5 | 352.5 |
| Evaluation | 243.46 | 263.07 | 275.24 | 291.22 | 319.19 | 326.01 | 337.86 | 350.19 |
| Target | 7.5 | 22.5 | 37.5 | 52.5 | 67.5 | 82.5 | 97.5 | 112.5 |
| Evaluation | 8.60 | 21.04 | 37.14 | 51.72 | 68.20 | 84.26 | 98.73 | 112.72 |
| Target | 127.5 | 142.5 | 157.5 | 172.5 | 187.5 | 202.5 | 217.5 | 232.5 |
| Evaluation | 129.20 | 152.40 | 157.35 | 308.21 | 189.11 | 201.22 | 218.09 | 230.26 |
| Target | 247.5 | 262.5 | 277.5 | 292.5 | 307.5 | 322.5 | 337.5 | 352.5 |
| Evaluation | 245.85 | 238.00 | 283.31 | 296.63 | 308.51 | 322.86 | 334.91 | 352.17 |

Table 18: 1:1 interpolation in Ant-Dir.

