# OpenReview forum: "Simple Emergent Action Representations from Multi-Task Policy Training"
_ICLR.cc/2023/Conference — ICLR 2023 poster_

### Official Review · Reviewer_YSuL · 2022-10-24

**Confidence:** 3
**Correctness:** 3
**Technical Novelty And Significance:** 3
**Empirical Novelty And Significance:** 4
**Recommendation:** 6

**Clarity, Quality, Novelty And Reproducibility:**

* Clarity: Fair. Suggestions for improvements on presentation are provided above.
* Quality: Good, though I think an ablation study on embedding normalization and noise injection could provide better understanding.
* Novelty: To the best of my knowledge, the findings presented in this paper are novel.
* Reproducibility: Fair. Sourcecode is not provided. But the appendix provides enough implementation details.

**Strength And Weaknesses:**

# Strength
* Overall the paper is clearly written and easy to follow. A few flaws undermines the readability a bit, which are discussed below.
* The finding of the emergence of the action embedding space is interesting. The proposed architecture is simple and straightforward. Yet this simple architecture yields interesting findings. The strong empirical performance against meta-RL baselines could inspire us to rethink the fast adaptation problem.

# Weakness
* The tasks in the experiments are relatively simple. Each task can be effectively described by one or two factors - which is nice for visualization and understanding. However, it is unclear if the same phenomenon would be observed in more complicated tasks such as object manipulation.
* I find the experiment results in Section 4.6 rather confusing. I do not understand the purpose of this experiment. When the task IDs are one-hot encoded, the input layer essentially serves as an embedding layer. It does change the interaction between the state embedding and the task embedding from concatenation to addition, but a certain equivalence can be shown for these two different types of interaction [1]. The fact that this one-hot variant does not even work well during training seems rather confusing to me. I will appreciate if the authors could provide better clarity on this experiment.
* One question I am interested in asking is the importance of embedding normalization and noise injection. I think it will be good to conduct an ablation study for this purpose.
* The section paragraph of the related work section looks confusing to me. The authors list a few works that address multi-task RL. But I don't see their connections to this work that are worth highlighting.
* Writing and presentation can be improved. For example, what are $\theta_{1}$, $\theta_{2}$, $\bar{\theta_{1}}$, $\bar{\theta_{2}}$ in Algorithm 1? The sampled action $a_{t}$ in the second for-loop is never used. Overall, the role of Algorithm 1 is unclear. It presents a standard multi-task policy training procedure which does not highlight LTE. Figure 3 and Figure 4 need more explanation in either the caption or the text. The screenshots are incomprehensible in their current form.

# Questions and discussion
* The second paragraph of Section 4.3 says "The coefficient $\beta$ is searched to better fit the target task". Why is search needed here? The new task is a convex combination of two reference tasks. I would expect we can just interpolate the embeddings with the same coefficient. Did the authors try it? What did you observe?

# Minor issues
* In the 5th line in the second paragraph on page 5: "The newly composed task is usually lie in a new modality..." Please fix the grammar.
* In the x-axis of Figure 5: what does "steps" mean? Does it mean episodes?
* In the second paragraph in Section 5: reference to Jeon et al and Chandak et al should use \citep{}.

**References**
1. Dumoulin _et al_ 2018, Feature-wise transformations, https://distill.pub/2018/feature-wise-transformations/

**Summary Of The Paper:**

This work studies the action representations emerged in multi-task policy training. The authors study a two-stream policy architecture for multi-task reinforcement learning. One input stream encodes the observation and the other input stream embeds the task ID. Finally, the two embedding vectors are concatenated and fed into the policy network. When proper normalization and noise injection are applied, the authors observed that the task ID embeddings space emerges where each embedding in this space encodes a meaning behavior. Embeddings in the emerged space can be interpolated or composed to generate interesting behaviors. The authors then further demonstrate that optimization in this embedding space can yield fast adaptation to downstream tasks. Visualization of the learned state embedding space and task embedding space provides more intuitive understanding of the learned structures.

**Summary Of The Review:**

Overall I think the findings in this paper are interesting and could benefit the community. But the presentation can be improved.

---

> ### Author Response · Authors · 2022-11-14
> **Response to Reviewer YSuL (Part 1/2)**
>
> Q1: *“The tasks in the experiments are relatively simple.”*
>
> A1: Our method can also be applied in the domain of manipulation. We add an experiment to train a robot arm to reach different points and perform fast adaptation with the policy. The results demonstrate that our method can still find suitable latent task embeddings in the action representation space for adaptation tasks in less than 5 steps. Please refer to the “Application in manipulation” section in the general response for further details.
>
> Q2: *“The fact that this one-hot variant does not even work well during training seems rather confusing to me. I will appreciate if the authors could provide better clarity on this experiment.”*
>
> A2: The goal to compare our method and the one-hot embedding is to show the design choices we found in the paper are crucial for discovering the action representations. Effectively, the interactions between sensory information and task information are both “concatenation” in EAR-SAC (our method) and OHE-SAC. However, in EAR-SAC, we first get LSEs and LTEs through MLP encoders and concatenate them together as the input of the action decoder, while in OHE-SAC, we concatenate the raw state and one-hot embedding at the beginning first and directly send them to an MLP-based policy network. Moreover, we add extra regularizations to the LTEs while the one-hot embedding does not. In an ideal world, these two approaches can be close to each other; however, in practice, we find that our implementation gives better representations, thus better results.
>
> Q3: *“I think it will be good to conduct an ablation study for this purpose.”*
>
> A3: We conduct an ablation study on the normalization and the injected noise. The results show that the normalization applied to action representations improves the quality and smoothness of the action representation space, while the injected noise reduces the sensitivity of the policy to perturbations in the environment and the agent itself. Please refer to the “Ablation Study” section in the general response for further details.
>
>
>
> Q4: *“The section paragraph of the related work section looks confusing to me. The authors list a few works that address multi-task RL. But I don't see their connections to this work that are worth highlighting.”*
>
> A4: In this work, we adopt the multi-task RL setting and train a multi-task policy network with a sensory-action representation layer. Specifically, multiple tasks share parameters of the state encoder and action decoder. Meanwhile, the action decoder is conditioned on LTE to tell which task the agent is faced with and thus avoid task conflicts. The purpose of this paragraph is to clarify the relation between multi-task RL methods and the training schemes that generate the action representations. We modify the multi-task RL section of the related work in our revised paper to clarify our statements.
>
>
>
> Q5: *“What are $\theta_1,\theta_2,\overline\theta_1,\overline\theta_2$ in Algorithm 1? The sampled action $a_t$ in the second for-loop is never used. ”*
>
> A5: We omitted some details in our algorithm for computing the SAC objectives and updating the SAC parameters. The sampled action $a_t$ is used to compute the policy objective of SAC defined in section 2, while the $\theta_1,\theta_2,\overline\theta_1,\overline\theta_2$ are parameters of the two soft Q-functions and their target networks defined in the Soft Actor-Critic algorithm[1].
>
> ### Reference:
>
> [1]Haarnoja T, Zhou A, Hartikainen K, et al. Soft actor-critic algorithms and applications[J]. arXiv preprint arXiv:1812.05905, 2018.

---

> ### Author Response · Authors · 2022-11-14
> **Response to Reviewer YSuL (Part 2/2)**
>
> Q6: *"Overall, the role of Algorithm 1 is unclear. It presents a standard multi-task policy training procedure which does not highlight LTE."*
>
> A6: In our method, we implement a multi-task policy network that is trained on multiple tasks. The key difference to a standard multi-task architecture is that we inject a sensory-action representation layer into the network. The sensory representation encodes raw state signals and the action representation is given by the LTE of the ongoing task. The sensory-action representation is then decoded into a motor action signal. In Algorithm 1, we describe the process of obtaining the representations described above in lines 12-14. LTE is injected with random noise and then normalized in line 13, sent to the action decoder as input in line 13, and updated together with other parameters of the policy network in line 18.
>
> Q7: *"Figure 3 and Figure 4 need more explanation in either the caption or the text. The screenshots are incomprehensible in their current form."*
>
> A7: For better presentation, we have updated Figure 3 and 4 in the revised paper. Figure 3 demonstrates the results of intra-action interpolation. We test three tasks interpolated between 1m/s and 2m/s and visualize the results. In all five tasks, the agent starts from the same point in the grid and runs for 100 steps. The figure shows the terminating state of each task. We find that, from the top row to the bottom row, the agent travels increasingly further, indicating that the velocity of the agent goes larger. Figure 4 demonstrates stop motion animations of the two composed tasks. Moreover, we provide animated results in our project website, please refer to https://sites.google.com/view/emergent-action-representation/
>
> Q8: *"The second paragraph of Section 4.3 says "The coefficient $\beta$ is searched to better fit the target task". Why is search needed here? The new task is a convex combination of two reference tasks. I would expect we can just interpolate the embeddings with the same coefficient. Did the authors try it? What did you observe?"*
>
> A8:In this paper, we choose a task-first philosophy: we first provide a desired task to the model and ask the model to accomplish this task by interpolation. To better fit the desired task, we perform a linear search during interpolating the action representations.However, we agree that observing how our method performs under a fixed coefficient can be an important and interesting way to analyze and understand our method. We conduct additional experiments in Appendix A.4. We fix the coefficient to be 0.5 and see whether the interpolated LTE can perform an interpolated task. The success rates of the two interpolation settings in all the environments are summarized in the table below.
>
> | Success Rate           | HalfCheetah-Vel | Hopper-Vel | Walker-Vel | Ant-Dir |
> | ---------------------- | --------------- | ---------- | ---------- | ------- |
> | Mid-task interpolation | 26/27           | 27/27      | 27/27      | 70/72   |
> | 1:1 interpolation      | 25/27           | 27/27      | 25/27      | 66/72   |
>
> Q9: *"In the x-axis of Figure 5: what does "steps" mean?"*
>
> A9: One “step” in our method means that we optimize the LTE once. A detailed description of the process of one “step” is demonstrated in Algorithm 2 in section 3.2, which is the same as an “adapt epoch” in line 3 of the algorithm.

---

> ### Author Response · Authors · 2022-12-08
> **Further Discussion**
>
> Dear reviewer YSuL,
>
> We would like to first thank you again for your constructive comments and helpful suggestions. Since we are nearly at the end of the discussion phase, we would like to post a follow-up discussion.
>
> In our previous response, we have clarified the raised questions and made corresponding improvements in the updated paper. We hope to further discuss with you whether your concerns have been addressed or not. If you still have any unclear parts of our work, please let us know.

---

### Official Review · Reviewer_VRfP · 2022-10-24

**Confidence:** 4
**Correctness:** 4
**Technical Novelty And Significance:** 2
**Empirical Novelty And Significance:** 3
**Recommendation:** 5

**Clarity, Quality, Novelty And Reproducibility:**

The paper is clearly written and would be straightforward to reproduce. I think the novelty of the approach is somewhat limited (basically a well-regularized task embedding) but there are some intriguing empirical results.

**Strength And Weaknesses:**

Strengths:
* interesting results with a relatively simple methods

Weaknesses:
* From what I can tell the paper proposes essentially a well-regularized version of learned task embeddings. It would be nice to see an ablation of the two constraints (noise and restriction to the unit sphere). Are both of these necessary to achieve the results shown? Would other regularization ideas work as well (e.g. information bottleneck?). It is interesting to see that this methods appears to perform well but it is unclear to me what specifically makes it work.
* relatively toy environments. I think the paper would be substantially stronger if the similar results were shown in more complex settings.
* I think task representation might be a better term than action representation (since action already has a specific meaning in RL). At the moment I think readers might be confused.

**Summary Of The Paper:**

This paper proposes an architectures that separately encodes sensory information and the task id in a multi-task setting. These are then concatenated to processed by a neural network that produces the action. The paper imposes constraints on the task representation (or action representation in the paper): the representation is restricted to the unit sphere and trained with some noise. The paper then evaluates the proposed architecture on a number of simple locomotion domains. Empirically the methods compares favourably to multi-task and meta-RL baselines. The paper also demonstrates that it is possible to interpolate in the task embedding space both within the same task (e.g. different velocities) and between different locomotion modes.

**Summary Of The Review:**

Interesting paper showing that a well regularized task embedding is competitive with multi task and meta-RL baselines on a set of relatively toy locomotion domains. The embedding also allows some interesting interpolation. I would like to see some more analysis to understand what specifically makes this approach work. I would also like to see if it scales to less toy settings.

---

> ### Author Response · Authors · 2022-11-14
> **Response to Reviewer VRfP**
>
> Q1: *"It would be nice to see an ablation of the two constraints (noise and restriction to the unit sphere). "*
>
> A1: We conduct an ablation study on the normalization and the injected noise. The results show that the normalization applied to action representations improves the quality and smoothness of the action representation space, while the injected noise reduces the sensitivity of the policy to perturbations in the environment and the agent itself. Please refer to the “Ablation Study” section in the general response for further details.
>
> Q2: *"I think the paper would be substantially stronger if the similar results were shown in more complex settings."*
>
> A2: Our method can also be applied in the domain of manipulation. We add an experiment to train a robot arm to reach different points and perform fast adaptation with the policy. The results demonstrate that our method can still find suitable latent task embeddings in the action representation space for adaptation tasks in less than 5 steps. Please refer to the “Application in manipulation” section in the general response for further details.
>
> Q3: *"I think task representation might be a better term than action representation."*
>
> A3: We thank the reviewer for the suggestion. In this paper, we view the sensory representations and the action representations as dual forms within a perception-decision system. As mentioned in the introduction, sensory representations reveal geometric and semantic structures hidden in the raw sensory signals. Symmetrically, we expect the representations of a similar structure to be developed to decode motor control signals. This new representation can be reused to quickly perform downstream tasks, regardless of exactly what the raw motor signals should be like. While we agree that the representations emerge as latent task embeddings, their core influences are observed in the process of decoding action sequences. Hence, we choose action representation as the terminology for this self-organizing space.
>
> Q4: *"I think the novelty of the approach is somewhat limited (basically a well-regularized task embedding)."*
>
> A4: The main contribution of this work lies in that we manage to learn a strong action representation space under a simple and straightforward network structure, which successfully performs different downstream tasks via zero-shot interpolation and composition as well as fast adaptation. It could be our future work to rethink the design of the policy network so that the idea in this paper can be applied to more complicated and real-world settings such as robotic control.

---

> ### Author Response · Authors · 2022-12-08
> **Further Discussion**
>
> Dear reviewer VRfP,
>
> We would like to first thank you again for your constructive comments and helpful suggestions. Since we are nearly at the end of the discussion phase, we would like to post a follow-up discussion.
>
> In our previous response, we have clarified the raised questions and made corresponding improvements in the updated paper. We hope to further discuss with you whether your concerns have been addressed or not. If you still have any unclear parts of our work, please let us know.

---

### Official Review · Reviewer_oCDb · 2022-10-25

**Confidence:** 3
**Correctness:** 2
**Technical Novelty And Significance:** 3
**Empirical Novelty And Significance:** 3
**Recommendation:** 5

**Clarity, Quality, Novelty And Reproducibility:**

The writing quality is moderate but can be improved through proofreading. The authors did a good job on presenting the ideas and experimental results.  The key idea is clear and intuitive, while implementation details need to be clarified. The proposed framework is novel to my knowledge.

**Strength And Weaknesses:**

## **[Strength]**

- The methodology is simple and intuitive (no additional loss functions for action abstraction).
- The idea and result are quite interesting.
- Interpolation/composition in the latent action space surprisingly works.
- The proposed framework provides novel insights on how to leverage the task distribution for learning high-level motor skills


## **[Weakness]**

### Related work is incomplete
While the paper mentioned Representation learning in RL, multi-task/metaRL in Sec. 5, important pieces are missing. A clearly related batch of studies are skill discovery with deep RL, which is closely related. Some related references are:
- Benjamin Eysenbach, Abhishek Gupta, Julian Ibarz, and Sergey Levine. Diversity is all you need: Learning skills without a reward function. In International Conference on Learning Representations, 2019
- Archit Sharma, Shixiang Gu, Sergey Levine, Vikash Kumar, and Karol Hausman. Dynamics-aware unsupervised discovery of skills. In International Conference on Learning Representations, 2020.
-  Kelvin Xu, Siddharth Verma, Chelsea Finn, and Sergey Levine. Continual learning of control primitives: Skill discovery via reset-games. In Advances in Neural Information Processing Systems, volume 33, 2020.

### The proposed framework heavily relies on the task set
To achieve the proposed results, there are quite a few requirements: e.g., a distribution of uni/multi-modal tasks with well-defined reward functions, the tasks shared the same state space. By contrast, skill discovery methods can achieve similar results without a reward function. Furthermore, people and animals can often self-develop motor primitives in a single task.

### Technical details are not clear
For example, model architecture, hyper-parameters, training schemes: Sec.A.3.2 is clearly not enough, not to say that source code was not provided. The authors should provide more details for reproducibility.

### Statistical significance of major claims is not well addressed
While the most interesting result --- interpretable interpolation and composition of LTEs, was well presented using an example agent (Figure 3, 4), it is unclear whether this result is common with different random seeds. That is to say, the authors did not demonstrate statistical reliability of the phenomenon showed in Figure 3,4, e.g., what is the percentage of random seeds that can achieve such a result?  More evaluations remain to be complete.

### Lack of ablation studies
Many design choices lack explanation. For example, the two techniques used to constrain the space, normalizing LTE and injecting random noise. Why normalizing LTE? What will happen if there is no such regularization? I guess these two techniques are essential to the success of the proposed framework, yet remains to be discussed.

### The writing quality is to be improved.
The paper seems not well polished. Belows are some issues I found. Nonetheless, I think the paper need a careful and throughout proofreading. Also, reference should also be manually checked.
- The first sentence in Sec.1: Deep RL can learn --> deep RL agents can learn
- Page 5: Specifically, We perform ... --> Specifically, we perform
- Page 5: We visualized the LSEs and and LTEs: duplicate "and"
- Page 9: the end, incorrect citation style
- Page 10: Duplicate citation of Chandak 2019
- Page 11: {Bayes}
- Page 14: control control


### **[Questions]**
1. Will the authors release source code?
2. How to initialize $\mathcal{Z}_\mathcal{T}$ ?
3. I am curious about the result of extrapolation in LTE space, e.g., if $\beta < 0$ or $\beta >1$, what will happen?



**Summary Of The Paper:**

The paper proposes a learning framework (Figure 1) for action abstraction in the scheme of multi-task reinforcement learning (RL). A latent action representation is assigned to each task, and the model tried to learn the tasks with state representation (shared among tasks) and action representations (specific to each task, task embedding) segregated. It was observed that the model self-developed interpretable and compositional action presentations via RL. The learned action representation can facilitate quick, gradient-free transfer learning. More interestingly, it was shown an interpolation in the latent action space led to an explainable "interpolation" in the behavior space. The main contribution is a proof of concept of a novel framework for action abstraction.

**Summary Of The Review:**

Overall, I appreciate this paper as it proposes a simple method to achieve interesting results, though it poses some demands to the task set. The goodness is the latent action representation showed self-organization via RL without additional, explicit loss functions for this purpose, which provides some insights for both machine learning and cognitive science about how motor primitives could be autonomously acquired and leveraged for transfer learning. The insights are, however, limited because there are a lot of points for the studies to improve as I mentioned above. Overall, my recommendation is around the borderline as for now, and my rating may increase given a constructive rebuttal.

---

> ### Author Response · Authors · 2022-11-14
> **Response to Reviewer oCDb (Part 1/2)**
>
> Q1: *“Related work is incomplete. While the paper mentioned Representation learning in RL, multi-task/metaRL in Sec. 5, important pieces are missing. A clearly related batch of studies are skill discovery with deep RL, which is closely related.”*
>
> A1: Thanks for pointing this out. We incorporate the literature of unsupervised skill discovery into the related work (Section 5). A more detailed discussion and comparison of the relationship between unsupervised skill discovery and our method are in the answer to the next question.
>
> Q2: *“The proposed framework heavily relies on the task set. To achieve the proposed results, there are quite a few requirements: e.g., a distribution of uni/multi-modal tasks with well-defined reward functions, the tasks shared the same state space. By contrast, skill discovery methods can achieve similar results without a reward function. ”*
>
> A2: While unsupervised skill discovery has achieved encouraging results, the results are rather different from what it proposed in this paper. We believe these two paradigms can be complementary and be merged as joint forces in the future. Here we discuss a few similarities and differences between them.
>
> While sharing the ability to develop different skills as that of unsupervised skill discovery, the learned representations in this work can be interpolated and re-composed to achieve new tasks with no extra training. We believe ultimately we will discover these representations with such manipulable properties in an unsupervised manner. However, none of the previous works prove this in the paper. Furthermore, both unsupervised skill discovery methods and our method can adapt to new tasks. However, the proposed method needs less than 3 optimization steps to achieve satisfactory performance in the face of new tasks while unsupervised approaches usually require abundant data sampling (~100k environment steps). These differences answer the reviewer’s question: the proposed method, although sacrificed to require more strict settings, can achieve task interpolation and composition in a zero-shot manner, and adapt to new tasks orders of magnitudes faster than the unsupervised approach. Hence, we conclude that the proposed approach and the unsupervised skill discovery methods are parallel yet potentially complementary efforts toward a similar goal. We believe that integrating the unsupervised methods into our architecture may also be a promising direction under exploration.
>
>
>
> Q3: *“Technical details are not clear. For example, model architecture, hyper-parameters, training schemes: Sec.A.3.2 is clearly not enough, not to say that source code was not provided. The authors should provide more details for reproducibility.”, “Will the authors release source code?”*
>
> A3: For better reproducibility, we add more implementation details including hyperparameters in Appendix A.3.2. We’ll release the official source code for this project soon; for reproducing our empirical results, please refer to the anonymous link: https://anonymous.4open.science/r/EAR-3EE7
>
> Q4: *"Statistical significance of major claims is not well addressed. While the most interesting result --- interpretable interpolation and composition of LTEs, was well presented using an example agent (Figure 3, 4), it is unclear whether this result is common with different random seeds. That is to say, the authors did not demonstrate statistical reliability of the phenomenon showed in Figure 3,4, e.g., what is the percentage of random seeds that can achieve such a result? More evaluations remain to be complete."*
>
> A4: We support the claims with experiments on multiple seeds. For interpolation, in most cases, mid-task interpolation and 1:1 task interpolation (defined in Appendix A.4) succeed. We summarize the success rates of interpolation on all seeds in the table below. For composition, composing walking and standing succeeds in all three seeds, while composing running and jumping only succeeds in two seeds. In the seed where the run-jumping task fails, we discover a new motion called move-jumping, in which the cheetah agent jumps backward with its hind leg raised. The additional quantitative results are added to Appendix A.4, 5.
>
> | Success Rate           | HalfCheetah-Vel | Hopper-Vel | Walker-Vel | Ant-Dir |
> | ---------------------- | --------------- | ---------- | ---------- | ------- |
> | Mid-task interpolation | 26/27           | 27/27      | 27/27      | 70/72   |
> | 1:1 interpolation      | 25/27           | 27/27      | 25/27      | 66/72   |

---

> ### Author Response · Authors · 2022-11-14
> **Response to Reviewer oCDb (Part 2/2)**
>
> Q5: *"Lack of ablation studies. Many design choices lack explanation. For example, the two techniques used to constrain the space, normalizing LTE and injecting random noise. Why normalizing LTE? What will happen if there is no such regularization? I guess these two techniques are essential to the success of the proposed framework, yet remains to be discussed."*
>
> A5: We conduct an ablation study on the normalization and the injected noise. The results show that the normalization applied to action representations improves the quality and smoothness of the action representation space, while the injected noise reduces the sensitivity of the policy to perturbations in the environment and the agent itself. Please refer to the “Ablation Study” section in the general response for further details.
>
> Q6: *"How to initialize LTE?"*
>
> A6: We initialize the LTEs of the training tasks with a 1-layer MLP task encoder. This encoder takes the one-hot encoding of different training tasks as input and outputs the corresponding LTE. We also add the description above to Section 3.1.
>
> Q7:*" I am curious about the result of extrapolation in LTE space, e.g., if $\beta < 0$ or $\beta >1$, what will happen?"*
>
> A7: We perform extrapolation between tasks in HalfCheetah-Vel and Ant-Dir. We find that, in HalfCheetah-Vel, the extrapolated behavior can also achieve the target speed as long as the speed is in the distribution of the training tasks. The exception happens when the target speed is out of the distribution (e.g. extrapolate Vel-1.0 and Vel-2.0 with $\beta >1$ or Vel-9.0 and Vel-10.0 with $\beta < 0$) where the agent can only perform extrapolation within a relatively small range of $\beta$. In the higher-dimensional environment Ant-Dir, extrapolation still works in most cases but has a higher failure rate. For more quantitative details on the extrapolation, please refer to Appendix A.4 in our revised paper.

---

> > ### Comment · Reviewer_oCDb · 2022-11-19
> > **Thanks for the reply**
> >
> > I appreciate the authors for the detailed response. While many of my concerns have been addressed. I would like spot a new concern.
> >
> > > A6: We initialize the LTEs of the training tasks with a 1-layer MLP task encoder. This encoder takes the one-hot encoding of different training tasks as input and outputs the corresponding LTE. We also add the description above to Section 3.1.
> >
> > I am disappointed the encoder takes the one-hot encoding of different training tasks as input, because this way of initialization provided strong inductive bias of task representation (i.e., the network know the identity of each task at the beginning). Therefore, the word **emergent** does not hold any more, and thus I cannot agree with the claim "*we present our finding that the task embeddings in a multi-task policy network can **automatically form a space** where action representations reside*". I conjecture that this way of initialiation is critical to the success of the proposed framework, and I am suprised that this important implementation detail did not appear in the original submitted version.
> >
> >
> > I would like to thank the authors for addressing other concerns, which lead to upgrading of my rating. However, given the above concern,  I decide to keep my current score.

---

> > > ### Author Response · Authors · 2022-11-19
> > > **Further Clarification**
> > >
> > > We thank the reviewer for responding to our reply and upgrading the rating! It is also encouraging that the reviewer acknowledged the other concerns were addressed!
> > >
> > > This remaining concern may be due to miscommunication. By "1-layer MLP task encoder", we mean a linear model without nonlinearity following random uniform initialization. The trivial one-hot vector selects the corresponding task embedding column from the task embeddings. This is precisely shown in Figure 1, and “Multi-Task Policy Training” is in the paper title. Interestingly, the randomly initialized task embeddings form self-organization and show generalization capabilities.
> > >
> > > In addition, we fully agree that it is not surprising that our multi-task framework can train the agent to learn multiple tasks. However, such a framework only supports the success of the training procedure but does not provide the effectiveness in our downstream settings, e.g., interpolation, composition, and adaptation tasks. When we directly use one-hot vectors to perform these tasks without the representations, it fails completely without error (please refer to Section 4.6 in the paper).
> > > We use the word **emergent** to describe **the representations in a self-organized space in which they can be reused, modified, and composed to perform downstream tasks**, but not simply those representations of the given training tasks.
> > >
> > > We hope to further discuss with you whether your new concern has been addressed or not. If you still have any unclear parts of our work, please let us know. Thanks.

---

> > > > ### Comment · Reviewer_oCDb · 2022-11-20
> > > > **Thanks for the reply**
> > > >
> > > > Thanks for the clarification. I did misunderstand the encoding of the LTE from the begining, but this was due to the lack of implementation details in the orginal submitted version (I checked again). I thought the task representation self-organized with totally randomly initiallized LTEs. After the authors' clarification that the task identity is needed for initializing LTEs, I find the results less amazing (while still interesting in terms of interpolation etc.) due to this inductive bias. I saw reviewer MYUq raising the same concern about the word **emergent**, which to me also seems an  over-claim.
> > > >
> > > > Therefore, I decides to maintain my orginial rating of borderline reject. Nonetheless, I would like to thank the authors for addressing my other concerns, e.g., providing extrapolation results.

---

### Official Review · Reviewer_MYUq · 2022-10-30

**Confidence:** 3
**Correctness:** 3
**Technical Novelty And Significance:** 3
**Empirical Novelty And Significance:** 3
**Recommendation:** 6

**Clarity, Quality, Novelty And Reproducibility:**

- The paper is relatively well-written and easy to follow, but more detailed invetigation on the detailed choice of action representation network would be useful.
- The approach looks technically sound in general, but details would be benefitical.
- The paper addresses a timely and important problem and presents a simple but new idea.


**Strength And Weaknesses:**

Strengths:
- This work is well-motivated. There is certainly a lack of effective task representation for complex multi-task environment. The proposed framework is addressing an timely and important problem.
- The proposed idea is interesting and looks making sense in general.
- The method is evaluated, compared agasint a reasonable choice of state-of-the-art

Weaknesses:
- The methodological details are not clearly described/analyzed. Perhaps the paper is to talk about the arguably simplea idea, but other than the framework in Figure 1, additional description/investigation is needed on what details in action representation would make impact on the performance. Would the MLP be the best choice for this purpose?
- The use of terminology is a bit abused. It is not very clear the task composition in this work can be called "emergent."


**Summary Of The Paper:**

This paper presents an action representation scheme for multi-task training, which is relatively simple in that the action representation is decoupled with the sensory represenation. The proposed method allows for emergent action composition as well as interpolation in the action space. The validity of the method is demonstrated on computational experiments in a Mujoco environment, compared against several other RL schemes.

**Summary Of The Review:**

The reviewer thinks that this paper presents an interesting idea on an important problem. The paper will be benefited by adding analysis on the detailed algorithm choices/variations.

---

> ### Author Response · Authors · 2022-11-14
> **Response to Reviewer MYUq**
>
> Q1: *"Additional description/investigation is needed on what details in action representation would make impact on the performance. Would the MLP be the best choice for this purpose?"*
>
> A1: To discover crucial detailed design choices, we conduct an ablation study on the normalization and the injected noise. The results show that the normalization applied to action representations improves the quality and smoothness of the action representation space, while the injected noise reduces the sensitivity of the policy to perturbations in the environment and the agent itself. Please refer to the “Ablation Study” section in the general response for further details.
>
> In terms of the choice of network architecture, we replace the first layer of MLP with an RNN layer to get the RNN encoder and decoder. Our MLP-based network is compared with the RNN-based network. We find that if we simply make such changes with a certain amount of tuning, our simple MLP-based network achieves much higher performance than an RNN-based one. We add this extra experiment in Appendix A.3.1.
>
> Q2: *"It is not very clear the task composition in this work can be called "emergent.""*
>
> A2: The action representations are emergent in the sense that the LTEs automatically self-organize into a geometrically and semantically meaningful structure during multi-task training without explicit restriction. When applied to downstream tasks, the self-organized LTEs capture the high-level semantics of a series of actions and serve as the action representations. In other words, “emergent” describes the organization process of the LTEs during training, while task composition is a setting to generate downstream tasks.

---

> ### Author Response · Authors · 2022-12-08
> **Further Discussion**
>
> Dear reviewer MYUq,
>
> We would like to first thank you again for your constructive comments and helpful suggestions. Since we are nearly at the end of the discussion phase, we would like to post a follow-up discussion.
>
> In our previous response, we have clarified the raised questions and made corresponding improvements in the updated paper. We hope to further discuss with you whether your concerns have been addressed or not. If you still have any unclear parts of our work, please let us know.

---

### Author Response · Authors · 2022-11-14
**General Response (Part 1/2)**

Dear reviewers, we appreciate all your feedback. We thank all the reviewers for considering that this work yields *“interesting results with a relatively simple method”*, and **Reviewer MYUq** and **Reviewer oCDb** for acknowledging the contributions of this work: *“The proposed framework is addressing a timely and important problem.”*, *“The proposed framework provides novel insights on how to leverage the task distribution for learning high-level motor skills.”*

In the following parts, we provide our response to general comments. We will address common concerns here and reply to each reviewer separately to address the remaining concerns. We welcome further discussion with each of the reviewers.

### Ablation Study

We ablate the two constraints (random noise injection and unit sphere regularization) mentioned in Section 3.1 of our method to better understand what makes it work in the proposed structure.

1. Random Noise Injection

We inject random noise into the LTEs during the training procedure to enhance the smoothness of the space. To assess the effect of this injected noise, we compare it with the case without the injected noise in the high-dimensional environment Ant-Dir. We observe that, when injected with random noise, our method achieves comparable training performance to the one without noise. Then we evaluate the policy we learn. For each task in each seed, we evaluate the trained policy for multiple trajectories. We find that without the injected noise in training, the policy will be sensitive to perturbation in the system during evaluation. For example, in the task of “direction-240” in Ant-Dir, we evaluate the policy for 5 trajectories and find that the second and third trials terminate before the maximal environment steps, which means they fail halfway. We count the total number of such failures respectively and summarize them in the table below. The policy trained without injected noise fails to perform over 20% of tasks stably, while the one trained with injected noise avoids such problems.

| Failure Rate  | Number of failure |
| ------------- | ----------------- |
| Without noise | 16 in 72 tasks    |
| With noise    | **0 in 72 tasks** |

2. Unit Sphere Regularization

We also apply regularization on LTEs by normalizing them on the surface of a unit sphere. We ablate this regularization by removing the normalization functions for LSEs and LTEs and directly concatenating them together as the input of the action decoder. We find that our method achieves higher sample efficiency and rewards than the one without normalization. Then we compare their performance in downstream tasks. We repeat the interpolation experiments described in Appendix A.4 with the new policy without normalization and calculate the success rates in the table below. With the normalization, the success rates of the interpolation experiments increase, which indicates that normalization improves the quality of the representation space and thus better supports interpolation. Furthermore, we also try to perform task adaptation with the new policy. Results show that the new policy can still adapt to some tasks while requiring more training steps.

| Success Rate          | Mid-task interpolation | 1:1 task interpolation |
| --------------------- | ---------------------- | ---------------------- |
| Without normalization | 64/72                  | 59/72                  |
| With normalization    | **70/72**              | **66/72**              |

For more quantitative details on the ablation study, please refer to Appendix A.6. We have added it to the revision of the paper.

### Application in Manipulation

In our paper, we mainly tackle the classic and widely studied locomotion problems in reinforcement learning. However, we add an experiment in the domain of manipulation to examine the effectiveness of the proposed method in other domains. We train a robot arm (based on the Metaworld benchmark) to reach different points and perform fast adaptation with the learned policy. Our method can still find suitable latent task embeddings in the action representation space for adaptation tasks in less than 5 steps. For more results of the multi-task training and adaptation procedures, please refer to Appendix A.7 in the revised paper.

---

### Author Response · Authors · 2022-11-14
**General Response (Part 2/2)**

We thank the reviewers for all the detailed comments and helpful suggestions. We have highlighted the changes in blue in the revised version of our paper. Here we provide an overview of our changes.

(1) In Section 3.1, we add the description of the initialization of LTEs of training tasks.

(2) In Section 4.3, we update Figure 3 and Figure as well as their captions.

(3) In Section 5, we update the paragraph of multitask RL part and add a new paragraph for unsupervised skill discovery.

(4) In Appendix A.3.1, we add the discussion on the choice of network architecture.

(5) In Appendix A.3.2, we update the hyperparameters for our method.

(6) In Appendix A.4, we add extra experiments and more quantitative results for intra-action interpolation.

(7) In Appendix A.5, we add more quantitative results for inter-action composition.

(8) In Appendix A.6, we add an ablation study on the two constraints we apply to the action representation space.

(9) In Appendix A.7, we add descriptions and experiments on applying our proposed method to the domain of manipulation.

---

### Decision · Program_Chairs · 2023-01-20

**Decision:**

Accept: poster

**Justification For Why Not Higher Score:**

The reviewers were not wholly convinced in the generality of the method.  The paper presents a "proof-of-concept", which merits additional validation in more general domains.

**Justification For Why Not Lower Score:**

The proposed method is simple and novel.  The presentation is clear enough for other researchers to replicate the work.  Even reviewers who were not entirely convinced in the generality of the proposed method thought it could inspire followup research.

**Metareview: Summary, Strengths And Weaknesses:**

Summary:
This paper develops the use of a latent space for actions across multiple reinforcement learning tasks. Tasks are represented with a one-hot encoding, and the latent space for actions is a low-dimensional sphere.  The experiments demonstrate interpretability and fast learning from the latent action space across multiple Mujoco locomotion tasks.

Strengths:
The reviewers found the work well-motivated (MYUq), simple (MYUq, oCDb, VRfp), with adequate experiments (MYUq) and interesting results (OCDb).  There was no major weaknesses in the results that were presented.

Weaknesses:
The experiments were adequate for the claims, but reviewers were unsure whether if similar results would hold for more complex domains (VRfP,YSuL,oCDb).  There were also concerns about the technical clarity of the paper, which were largely resolved after the author response.


**Note From Pc:**

if the above contains the word "oral" or "spotlight" please see: "oral" presentation means -> notable-top-5% and "spotlight" means -> notable-top-25%. As stated in our emails, we are disassociating presentation type from AC recommendations